# Nexus between carbon emissions, energy consumption, and economic growth: Evidence from global economies

**Hasara Dissanayake[1], Nishitha Perera[1], Sajani Abeykoon[1], Diruni Samson[1], Ruwan Jayathilaka[2]\*, Maneka Jayasinghe[3], Shanta Yapa[1]**

**1** SLIIT Business School, Sri Lanka Institute of Information Technology, Malabe, Sri Lanka, **2** Department of Information Management, SLIIT Business School, Sri Lanka Institute of Information Technology, Malabe, Sri Lanka, **3** Business and Accounting, Senior Lecturer, Faculty of Arts and Society, Charles Darwin University, Waterfront Campus, Darwin, Australia

\* ruwan.j@sliit.lk

**Data Availability Statement:** The data underlying the results presented in the study are available

## Abstract

Renewable energy holds a remarkable role in clean energy adaptation due to the much lower carbon footprint it releases compared to other fossil fuels. It also has a positive impact by slowing down the rate of climate change. The study has examined the links between renewable and non-renewable energy use, $CO_2$ emissions and economic growth in developed, developing, and LDCs and Economies in Transition between 1990 and 2019 in 152 countries. Granger-causality has been used as the methodology to investigate the link between the variables. The findings of the existing studies on the relationship between the consumption of renewable and non-renewable energy sources and economic growth are inconsistent, indicating that there may or may not be a relationship between the two factors. Apart from having a few empirical studies so far have examined the link between the above-mentioned variables, analysis has yet to encompass all the regions in the four sub-groups discussed above. The results indicated that no Granger-causal relationship exists between GDP and REC outside of Economies in Transition. Additionally, the GDP and $CO_2$ of all countries have a one-way relationship. Nevertheless, research indicates that GDP and $CO_2$ have a bi-directional link in Economies in Transition, a uni-directional relationship in developing countries, and no meaningful association in developed and LDCs. Therefore, it is essential to emphasise actions to lower $CO_2$ emissions and develop renewable energy while also stimulating the economy. Ultimately, more nations should choose renewable energy sources to build a more sustainable future.

## Introduction

Carbon Dioxide ($CO_2$) emissions have made it hazardous for the globe to consume non-renewable energy [1]. Burning coal and oil to generate electricity releases heat-trapping gases into the atmosphere that contribute to global warming and climate change. The positive relationship between Non-renewable Energy Consumption (NREC) and $CO_2$ emissions is well established

from World Bank Indicators (https://data.worldbank.org/indicator) datasets.

**Funding:** The authors received no specific funding for this work.

**Competing interests:** The authors have declared that no competing interests exist.

in the literature [2]. For instance, the Intergovernmental Panel on Climate Change's most recent study note that if emissions are not kept in check by 2040, global warming might surpass 2.7 degrees Fahrenheit beyond pre-industrial levels, posing a grave threat to life [3]. Hence, renewable energy would be the ideal replacement as it can lower $CO_2$ emissions. With more innovative technology on the market, there are less costly ways to capture and retain wind and solar energy. As such, there is an increasing tendency to adapt more renewable energy sources in countries across the world. For example, American households and businesses switched to 100% pollution-free power which provides services to nearly 83 million households and businesses. This represents about half of the electricity market of the United States of America (USA) in 2019 and has led to much more production of wind and solar [4].

However, clean energy transition is not only costly and time consuming but also requires substantial government-led capital and investment strategy. Hence the developing countries face substantial challenges in transitioning to renewable energy sources [5]. The analysis done by some researchers found that renders space for making policy suggestions as government spending on research and development is essential to achieving the goal of investing in renewable energy by all nations [6]. It is essential to create emission-reducing technologies and implement strict environmental regulations in order to reduce ecological footprint [6]. Similarly, the government should adopt an effective and sustainable approach in ensuring energy consumption optimization and economic growth [7]. Furthermore, it has been found that energy consumption results in greater emissions in both short and long term globally [8]. In addition, the scientific literature provides mixed findings on the nexus between Renewable Energy Consumption (REC) and economic growth, owing to different methodological approaches, types of data used and time period under consideration. For example, Gross Domestic Product (GDP) per capita, renewable energy, and non-renewable energy have different effects on $CO_2$ emissions in BRICST (Brazil, Russia, India, China, South Africa and Turkey) countries [9]. Similarly, some studies find no two-way causal relationship [10–12], while some other studies find evidence of two-way causal relationship between REC and economic growth [13–17]. Besides, some suggest that there is a uni-directional causality running from GDP to REC [18, 19].

The objective of this study is to examine the correlations between energy consumption (renewable and non-renewable energy), $CO_2$ emissions, and economic growth over the period 1990–2019 in 152 nations across the globe. More specifically, this study identifies the differences in the observed relationships across four different country groups: 29 developed, 72 developing, 13 transitional economies (Albania, Armenia, Azerbaijan, Belarus, Georgia, Kazakhstan, Kyrgyz Republic, North Macedonia, Russian Federation, Tajikistan, Turkmenistan, Ukraine, Uzbekistan) and 38 least-developed countries (LDCs). This study is primarily motivated by the lack of consensus on the precise relationship between energy consumption (renewable and non-renewable), $CO_2$ emissions and economic growth at a global scale that enables meaningful comparisons across different countries.

To this end, the present research contributes to the knowledge base and endeavours to fill gaps in the existing literature in 4 ways; (1) although economic growth, non-renewable energy, and renewable energy nexus has been well-researched in the literature, very little research has been carried out on LDCs or transitional economies. Since these countries are highly dependent on fossil fuel for economic activities, empirical evidence from these countries will provide important insights for policymakers and our research contributes to this limited body of literature; (2) this study extends the knowledge base on this important research area by using a variety of techniques such as violine graphs, panel Granger-causality test, and the impulse-response functions (IRF) plots which have not widely used in the existing research on this topic. Hence, this study provides a further nuanced understanding on the link between energy

consumption, $CO_2$ emissions, and economic growth. To ensure the reliability of the findings, before carrying out the Granger-causality test, the Panel Vector Auto-Regression (pVAR) was checked to see if it is satisfied or failed to satisfy the stability criteria for all variables for all countries under consideration; and (3) the majority of existing studies focus either on single country or a small group of countries. Such research provides limited possibilities for comparison across countries with different stages of economic development. This study carries out a global level analysis as well as a disaggregated analysis by developed, developing, LDCs, and transitional economies, enabling a comparison across countries in different stages of economic development; and (4) current study covers a wide time period (1990–2019) and uses the most recent for the empirical analysis. We believe that a comprehensive study at global level, like ours, would enhance the evidence base on this important area of concern to facilitate decision-making. However, this study does not address the creation of jobs, improved quality of life in rural areas, improved public health as a result of reduced pollution, or increased public and professional awareness; it is because the study focused only on renewable energy, environmental sustainability and economic growth.

The remainder of the research is structured as follows: In the next section, a review of literature on the links between energy consumption, economic growth, and $CO_2$ emissions is presented followed by the data and methodology section, which presents the econometric model used for the analysis of Granger-causality, and results and discussion section. The last section presents concluding comments.

## Significance of the study

This study classifies nations as developed, developing, transitional economies, and LDCs using data from the World Bank for the years between 1990 and 2019 for 152 countries to determine the relationship between global GDP growth, energy consumption and carbon footprint. The management and use of renewable energy sources are crucial due to the global nature of the energy challenges. Renewable energy is pure, secure, and unbounded, in contrast to conventional energy. Therefore, it is rapidly acquiring momentum worldwide, and as predicted, will soon displace many traditional energy sources and take the lead concerning the overall proportion of energy consumption.

Developing and LDCs continue to experience rapid, unsustainable economic growth. However, by switching to renewable energy, sustainable growth can also be experienced. Using renewable technologies can benefit local power generation in rural and remote areas. Similarly, lowering carbon footprint is essential for preventing climate change, improving sustainable development, generating financial savings, and upholding international commitments. It is a shared obligation that seeks participation and collaboration from all key actors including individuals, groups, and governments worldwide.

## Theoretical framework

The relationship between REC, NREC, economic growth, and $CO_2$ emissions is explained by 3 theoretical frameworks namely, (1) the 3Ps framework, (2) Natural Resource Curse and (3) the Porter Hypothesis. (1) The 3Ps framework is a common concept in the disciplines of sustainability and corporate social responsibility. The 3Ps stand for People, Planet, and Profit. These three factors must be taken into consideration by businesses while implementing sustainable practices. Planet and profit from the aforementioned three factors have been used in this study. Due to their contribution to reduced greenhouse gas emissions and lower detrimental effects on natural ecosystems, renewable energy sources have a lower environmental impact than conventional fossil fuels. Lowering $CO_2$ emissions is crucial for decreasing effects of

climate change, such as sea level rise, more frequent and severe weather, and other detrimental effects on the ecosystem. These components fall within the concept of 'planet'. Long term energy cost savings, improved brand reputation, and access to new markets that value sustainability can all be achieved by businesses using renewable energy. Economic growth can also help businesses by raising consumer demand for goods and services, but to ensure sustainability over the long term, it must be in harmony with sustainability considerations. These elements can be considered as the 'profit' [20]. (2) Natural resource curse is the theory which highlights the difficulties that countries confront in managing their natural resources. The development of renewable energy sources also offers a theoretical framework for tackling these issues and fostering economic growth. Diversifying the economy to minimise the reliance on natural resources is one strategy for overcoming the resource curse. By investing in energy alternatives that don't rely on non-renewable resources, renewable energy gives nations the opportunity to accomplish these objectives. This can also reduce dependency on fossil fuels exports and result in the development of new businesses and jobs [21]. According to (3) the Porter Hypothesis, environmental laws can encourage innovation, competitiveness, and economic growth. This concept suggests that environmental rules encourage companies to develop more effective and environmentally friendly production methods by opening avenues for new markets for clean technologies. For businesses that use cleaner technologies, this enables cost savings, better profitability, and increased competitiveness. According to the Porter Hypothesis, environmental restrictions can open up new markets for renewable energy technology, lowering reliance on non-renewable energy sources and preventing climate change. This is relevant to renewable energy. Environmental laws can also boost the economy by enhancing public health, decreasing healthcare expenses, and boosting productivity [22]. This study analysed the estimation results in the context of these well-accepted theories to illustrate the link between economic growth, energy consumption and $CO_2$ emissions.

## Literature review

In order to carry out this study, a thorough review of the literature was conducted, focusing solely on the effects of renewable energy, non-renewable energy, and $CO_2$ emissions on economic growth concerning 4 country groups. The Fig 1 presents the systematic approach adopted for selecting the research papers for the literature review. As elaborated in figure, 152 articles were found using keywords and database searches and 83 publications were removed. The remaining 69 papers were divided into four categories: developed countries, developing countries, transitional economies, and LDCs.

### Developed countries

When considering the literature on REC and economic growth, the findings suggest no consensus. While some research find that there is a uni-directional relationship running from economic growth to REC and *vice versa*, some other research show that there is a bi-directional relationship between the two variables. An in-depth review of literature revealed that the majority of the existing research on the link between REC and economic growth focuses on industrialized nations. Some studies have identified bi-directional relationships between REC and GDP growth by utilizing Granger-causality and wavelet analysis methods, descriptive statistics, cluster analysis and Auto-Regressive Distributed Lag (ARDL), respectively in European Union member countries [14, 16, 17]. Some researchers have found a long-run uni-directional causality running from REC to economic growth in Germany, Italy, and the United Kingdom (UK), a short-run uni-directional causality running from economic growth to REC in Italy and the UK, and a long-run uni-directional causality running from economic growth to REC

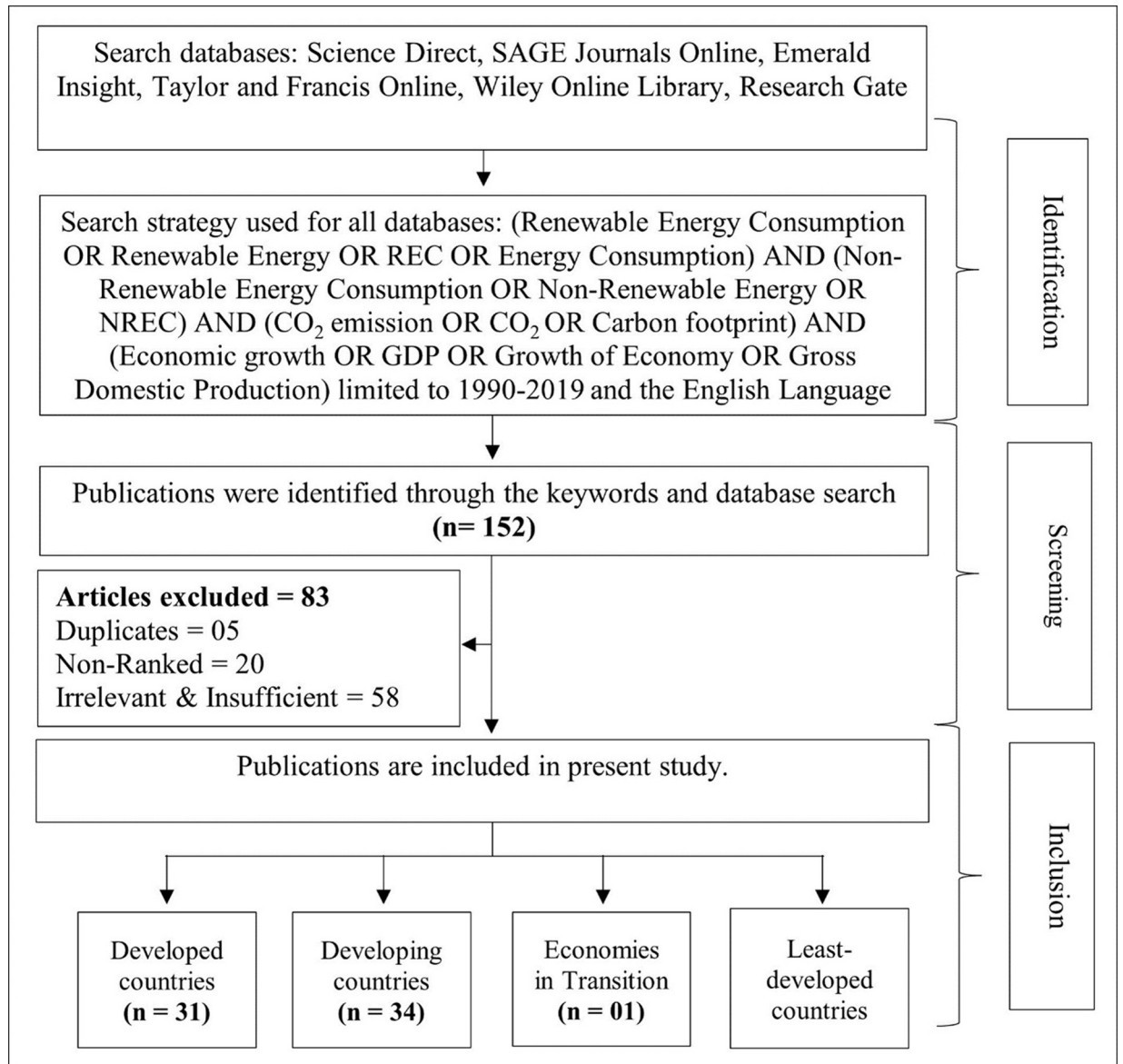

**Fig 1. Flow diagram for the review.** Source: Authors' Composition.

in the USA [18, 19]. A similar long-run uni-directional causal relationship from economic growth to REC was identified in Canada and the USA [23]. For the Granger-causality inference, a uni-directional causal interaction from economic growth to $CO_2$ emissions is found in the top 10 energy transition economies (China, USA, UK, Germany, France, Japan, India, Spain, Brazil, and South Korea) [24]. The Dumitrescu and Hurlin causality test revealed a uni-directional causal relationship between economic growth and energy consumption in the top 5 carbon emitter nations (China, USA, India, Brazil, and Russia) [25]. Additionally, economic activities are impacted by economic policy uncertainty (EPU), which can result in significant $CO_2$ emissions [26]. The uni-directional causality from real income to energy has been identified through causality analysis in Italy [27], demonstrating that continuous economic

expansion also results in a constant increase in energy consumption. The study covered a long time period from 1960 to 2014.

Similar to the lack of unanimity in the direction of the casual relationship between GDP and REC, there is no consensus on the direction (negative or positive) of the relationship in the literature. For example, while some studies have concluded that REC harms economic growth [28, 29], some other studies suggested that there are favourable relationships between REC and economic growth [13, 30, 31]. Furthermore, a few studies have shown the interrelation between monetary policy to economic growth and energy consumption. As per the research conducted in the USA using the bootstrap ARDL method, energy consumption negatively impacts monetary policy uncertainty between 1990M1 and 2020M12 [32]. By utilizing the unit root test, it turned out to be that expansionary monetary policy positively impacted the REC during the long-run, and short-run, and inversely in the USA [33]. Similarly, in another study based on the USA, the dynamic ARDL model's findings show that while fiscal policy uncertainty reduces emissions, monetary policy uncertainty increases emissions for both short and long- run. Last but not least, trade policy uncertainty has no impact on $CO_2$ emissions [34]. Furthermore, according to the bootstrap ARDL technique results in the USA, excessive EPU may be accountable for short term environmental degradation because it increases $CO_2$ emissions. In contrast, EPU reduces $CO_2$ emissions in the long-run, suggesting that high EPU improves environmental quality in the long-run [35]. The empirical results of a research conducted for the Nordic nations (Norway, Denmark, Finland, Sweden, Iceland, Greenland, land, and the Faroe Islands) show that economic growth reduces $CO_2$ emissions both in the short and long-term [36]. Furthermore, energy use and income level are found to enhance $CO_2$ emissions in Iceland using fully modified ordinary least squares regression and Dynamic Ordinary Least Square regression [37]. A research suggested that in countries with high GDP per capita the renewable energy will account for a more significant portion of final consumption [38]. The study investigated the implications of REC on the cost of fossil fuels [39]. The results revealed that the substitution between renewable and non-renewable energy sources is supported by the short and long-term causality from coal and natural gas pricing to REC. Additionally, the same study found no significant Granger-causal relationship between the use of renewable energy and the cost of crude oil [39]. Some studies, however, found no significant relationship between GDP and REC. For example, based on a panel estimation techniques found no any relationship between REC and economic growth in Scandinavian countries [40]. Nevertheless, overall, there is an overwhelming support for using renewable energy as a part of innovative energy policy roadmaps to expedite the ecological shift.

With respect to the relationship between economic growth and $CO_2$ emissions, a study based on the wavelet-based Granger-causality approach revealed a negative correlation between economic growth and $CO_2$ emissions in Sweden [41]. Relatedly, some revealed that the renewable energy utilisation dramatically reduces $CO_2$ emissions, whereas sustainable economic expansion increases $CO_2$ emissions [42]. This study further revealed a bi-directional causality running from REC to $CO_2$ emissions, technological innovation to $CO_2$ emissions, GDP to REC, and REC to technological innovation. According to the USA-based studies, energy efficiency reduces $CO_2$ emissions both in the long and short-run. Additionally, the novel Fourier ARDL model shows that both in long and short-run, economic growth and population increase $CO_2$ emissions [43]. In the long-run, it has been identified that globalization, renewable energy, and economic complexity reduce carbon emissions in the top 10 energy transition economies (Iceland, France, New Zealand, UK, Finland, Austria, Switzerland, Denmark, Norway, and Sweden) [44]. A study based in countries in the Organization for Economic Cooperation and Development (OECD) found that a positive relationship between $CO_2$ emission and fossil fuel consumption where a 1% increase in fossil fuel consumption will

increase $CO_2$ emission by 0.76%, and a negative relationship with REC where a 1% increase in REC will reduce $CO_2$ emission by 0.14% [45].

Additionally, some revealed that while economic expansion lessens the ecological footprint, democracy and environmental legislation favour ecological sustainability by increasing the ecological footprint [46]. Some studies' primary suggestions include decreasing fossil fuel subsidies and increasing investment in renewable energy [47]. This research suggested potential positive or negative potential association between REC, NREC, and $CO_2$ emissions as well as possibilities for presence of no significant relationship.

## Developing countries

Considering the empirical findings on the relationship between REC and economic growth in the context of developing countries, it has been observed that there is a bi-directional causal relationship between running from REC to economic growth in a group of developing countries, including China [15, 48]. Similarly, based on an analysis of Granger-causality test, a bi-directional relationship has been found between NREC and economic growth in African Organisation of the Petroleum Exporting Countries (OPEC) member countries [49]. Additionally, a study on Brazil, Russia, India, China, and South Africa (BRICS) countries found that the regressors of technological innovation, globalization, non-renewable energy, and economic expansion have a bi-directional causal relationship with ecological footprint, according to the Dumitrescu-Hurlin panel causality test [50]. Based on an analysis of 42 developing countries, the study found a long-run uni-directional causal relationship running from REC to economic development [51]. Moreover, research conducted for the BRICS nations have noticed a one-way causality from economic complexity, non-renewable energy, economic growth, renewable energy, and financial risk to ecological sustainability [6]. On the one hand, a study conducted for the E7 countries (Brazil, China, India, Indonesia, Mexico, Russia, and Turkey) between 1990 and 2015 proved that using renewable energy improves environmental quality by reducing both ecological footprint and $CO_2$ emissions. On the other hand, using non-renewable energy increases both the environmental impact and $CO_2$ emissions. The results of the causality test suggest that there is a uni-directional causal relationship running from NREC to economic growth [52]. Furthermore, it has been found that there is a uni-directional causality running from GDP to energy consumption in the short-run as well as long-run in Bangladesh [45].

In terms of the direction of the relationship, a study based on nonparametric panel modelling approach, revealed that increase in REC would lead to a higher economic growth, indicating a positive relationship between the two variables in the context of 11 developing countries across different regions [53]. In Africa, a positive relationship between economic growth and REC has been identified in the long-run. However, the same study found no significant relationship between the environmental sustainability, $CO_2$ and economic growth [54]. The use of renewable energy and globalisation have been shown to reduce emissions in Argentina, whereas the use of non-renewable energy has been found to increase emissions over the short and long-term according to a study based on the ARDL model [55]. Similarly, in China, $CO_2$ emissions are positively influenced by economic growth, coal consumption, and natural resources at all frequencies, or both in the short and long-run, leading to environmental degradation, using the Wavelet Local Multiple Correlation bivariate cases [56]. According to the study that utilised quantile-on-quantile regression (QQR) method, Nigeria and Mexico experienced positive effects of energy resources on economic growth in the majority of quantiles [57]. The positive impact of economic growth and non-renewable energy on renewable energy usage has been confirmed by research done in BRICST economies using panel quantile

regressions [58]. Contrarily, the consumption of renewable energy is negatively impacted by the $CO_2$ emissions [58]. Another study on BRICS economies revealed that the use of renewable energy, exports, and technological advancements reduce consumption-based carbon emissions ($CCO_2$), whereas economic expansion and imports increase $CCO_2$ [59]. Similarly, another study has found that there is a positive and significant relationship between NREC and economic growth in BRICS countries [60]. However, the positive relationship between REC and economic growth was not significant. On the other hand it was shown that both REC and NREC are positively correlated with economic growth in 39 developing countries [61].

Moreover, it was also found that there is a negative relationship between REC and economic growth in South Africa, west African countries, Ghana, and 16 Asian countries, respectively [5, 62–64]. Likewise, the results of the QQR show negative effects on economic expansion in the majority of the quantiles in Indonesia and Turkey [57]. Some studies, however, concluded that no significant relationship exists between REC and NREC and economic growth [10, 12, 65]. Relatedly, using various linear panel model techniques, some studies have showed that there is no significant relationship between REC and economic growth in developing and non-OECD countries [66].

Similarly, some studies have revealed a few variations in the relationship between REC, economic growth and $CO_2$ emissions in the long-run and short-run in Iran [39]. In particular, the study revealed that there is no significant (negative) impact of reducing $CO_2$ emissions and REC on economic growth in the short-run. However, in the long-run, increase in $CO_2$ emissions and REC leads to an increase in economic growth. A study based on Thailand time-series data from 1971 to 2013 revealed that an increase in the NREC would result in higher $CO_2$ emissions and threatens the environment [2]. Similar results were found in Vietnam-based study utilising the wavelet analysis [67]. Furthermore, in N11 countries, it was revealed that the clean energy consumption and a higher per capita income positively impact environment [68]. Using the Nonlinear Auto-Regressive Distributed Lag (NARDL) method, some research revealed that NREC had a significant negative impact on economic growth in Pakistan during 1970–2018 [69]. Similarly, it has been revealed that renewable energy stimulates economic growth, and labour and capital positively and significantly impact GDP [70]. Examining the factors affecting REC, [44] suggested that GDP and institutional factors affect the level of REC across 41 European countries, while the environmental component only has a negligible impact on non-extractive countries. Moreover, some studies have found that other factors, such as trade openness, appear to have a statistically significant impact on REC [71].

## Transitional economies

A review of literature on the nexus between REC, NREC, $CO_2$ emissions and economic growth revealed that, to the best of authors' knowledge, there is only one study, on transitional economies. Using a variety of time-series modelling techniques, the study examined the nature of short-term and long-term relationships between REC, economic growth and financial development in Russia over the period 1990 to 2014 [11]. The result of Granger-causality test showed that REC does not Granger cause economic growth or financial development. Granger-causality However, economic growth Granger causes changes in REC. Moreover, the result of the Granger-causality test suggested the presence of bi-directional relationship between economic growth and financial development during the period under consideration.

## Least-developed countries

Research on the link between energy consumption, $CO_2$ emissions and economic growth is scarce in the context of LDCs. The NARDL test results based on Rwanda showed that REC has

a positive impact on economic growth [72]. Moreover, using the Granger-causality and mediation models, study based on Ghana found that REC has a direct and considerable impact on economic growth [48]. Examining the determinants of REC in 32 African countries, [73] have identified that financial freedom and well-being as the primary factors influencing the share of renewable energy in overall energy consumption.

In summary, although there is a significant body of literature examining the link between REC, NREC, $CO_2$ emissions and economic growth, there is no consensus on the causal relationship and the direction of the relationship between the variables. Significant research gaps also exist. In particular, there is an evident lack of empirical research for the aforementioned 4 country categories based on national income. Also, there were limited research available in the context of transitional economies and LDCs. The current study contributes to this gap in the existing literature.

## The data and methodology

This section discusses the data sources and the statistical models employed in this study.

### The data

The objective of this study is to examine the nexus between REC, NREC, $CO_2$ emissions and economic growth in 152 countries between 1990 and 2019. Data for the empirical analysis is sourced from the World Bank database as detailed in Table 1. The data file used for the study is presented in the S1 Appendix.

Following the existing literature, the annual GDP growth rate was used as a measurement of economic growth. REC is defined as the renewable energy consumption as percentage of total final energy consumption; NREC is calculated as 100%-REC; $CO_2$ emissions is metric tons per capita. For the countries with missing annual data, the researchers used forecasted values based on the average values of the nearest five years. The Marshall Islands, Micronesia Federal States, the United Arab Emirates in 1990 and 1991, Namibia in 1990, North America in 2016, 2017, 2018, and 2019 have forecasted values for REC and NREC. Forecasts were made for the $CO_2$ emissions in Namibia in 1990, Micronesia Federal States, the Marshall Islands in 1990 and 1991, and Mali in 1996, 1998, and 1999. Azerbaijan, Belarus, Kazakhstan, Armenia, Yemen Republic, Slovak Republic, North Macedonia, Poland, Romania, and Greece all lacked GDP growth in 1990. Additionally, the GDP growth rates for Syria Arab Republic, the Slovak Republic, Hungary, and Kiribati were missing in the years 2019, 1991, 1992, 1991, 1991, and 1995.

**Table 1. Data and source of the parameters.**

| Variable | Measurement unit | Source |
|---|---|---|
| Renewable energy consumption | % of total final energy consumption | World Development Indicator https://data.worldbank.org/indicator/EG.FEC.RNEW.ZS |
| Non-renewable energy consumption | 100 - % of total final energy consumption | World Development Indicator https://data.worldbank.org/indicator/EG.FEC.RNEW.ZS |
| $CO_2$ emission | Metric tons per capita | World Development Indicator https://data.worldbank.org/indicator/EN.ATM.CO2E.PC |
| Economic Growth | GDP annual % | World Development Indicator https://data.worldbank.org/indicator/NY.GDP.MKTP.KD.ZG |

Source: Authors' Compilation.

## The model

There are various approaches to run the Granger-causality [74] tests in panel data models. This study employs the method which treats the autoregressive coefficients and regression coefficient slopes as constants [75–77]. In this study, the causality between REC, NREC, $CO_2$ and economic growth has been determined using the Panel Granger-causality test. Eq 1 was used to assess causality and its direction [78]. Then, the Granger-causality has also been applied using the pVAR model. We also carried out Levin-Lin-Chu (LLC) [79–81] unit root test to determine whether the data sequence is stationary. The stability test was carried out to assess the stability of the variables [82–84]. Eq 1 will be used to assess causality and the direction of the relationship.

$$Y_{i,t} = \sum_{k=1}^{\rho} \beta_i \, Y_{i,t-k} + \sum_{k=0}^{\rho} \theta_k \, X_{i,t-k} + u_{i,t} \tag{1}$$

Y is the dependent variable (area and duration are represented by i and t, respectively), X is the independent variable, $u_{i,t}$ is the error term, and k is the frequency of lags. U is normally distributed with $u_{i,t} = \alpha_i + \varepsilon_{i,t}$, $\rho$ is the number of lags, and $\varepsilon_{i,t}$, are $i.i.d.$ $(0, \sigma^2)$. Since the variables REC, NREC, $CO_2$, and economic growth have been linked in this study, it is impossible to categorize them precisely as dependent and independent variables. For instance, in Granger-causality test, REC was used as both a dependent and independent variable. The study also employed the difference of variables to eliminate lags in the model in Eqs 2 to 7. The estimated models were as follows:

$$DGDP_{i,t} = \sum_{k=1}^{\rho} \beta_i \, DGDP_{i,t-k} + \sum_{k=0}^{\rho} \theta_k \, DREC_{i,t-k} + q_{i,t} \tag{2}$$

$$DREC_{i,t} = \sum_{k=1}^{\rho} \gamma_i \, DREC_{i,t-k} + \sum_{k=0}^{\rho} \pi_k \, DGDP_{i,t-k} + s_{i,t} \tag{3}$$

$$DGDP_{i,t} = \sum_{k=1}^{\rho} \delta_i \, DGDP_{i,t-k} + \sum_{k=0}^{\rho} \rho_k \, DNREC_{i,t-k} + u_{i,t} \tag{4}$$

$$DNREC_{i,t} = \sum_{k=1}^{\rho} \Omega_i \, DREC_{i,t-k} + \sum_{k=0}^{\rho} \sigma_k DGDP_{i,t-k} + v_{i,t} \tag{5}$$

$$DGDP_{i,t} = \sum_{k=1}^{\rho} \vartheta_i \, DGDP_{i,t-k} + \sum_{k=0}^{\rho} \tau_k \, DCO_{2i,t-k} + w_{i,t} \tag{6}$$

$$DCO_{2_{i,t}} = \sum_{k=1}^{\rho} \lambda_i \, CO_{2i,t-k} + \sum_{k=0}^{\rho} \omega_k \, DGDP_{i,t-k} + z_{i,t} \tag{7}$$

where DGDP, DREC, DNREC, and $DCO_2$ are the first difference of GDP, REC, NREC and $CO_2$, respectively. $\beta_i$, $\gamma_i$, $\delta_i$, $\Omega_i$, $\vartheta_i$, and $\lambda_i$ are regression coefficients, where $\theta_k$, $\pi_k$, $\rho_k$, $\sigma_k$, $\tau_k$, and $\omega_k$ are constant for $k \in [1,N]$. $Q_{i,t}$, $s_{i,t}$, $u_{i,t}$, $v_{i,t}$, $w_{i,t}$, and $z_{i,t}$ are the error terms and assumed to satisfy the standard properties, i.e., they are independently, identically, and normally

distributed, and free from heteroskedasticity and autocorrelation. The models were estimated using STATA version 12.

In the field of causal analysis, the use of Granger-causality offers several benefits and insights. Firstly, it allows us to identify causal interference, which involves examining whether one time series can effectively forecast another. By establishing such causal relationships, the accuracy of forecasting processes can be significantly enhanced, leading to more reliable predictions. Secondly, Granger-causality aids in determining the direction of causality. This is crucial for researchers as it helps identify which variable serves as the leading factor and which one lags behind in a causal relationship. This understanding of directionality enables the identification of key variables and their relationships, providing valuable insights for both researchers and policy makers in formulating appropriate policies. Furthermore, Granger-causality analysis provides a measure of the strength of the causal relationship at selected significance levels. This information empowers researchers and policy makers to evaluate and assess the reliability of the obtained results before making informed decisions and taking appropriate actions. Therefore, it can be assumed that this study using Granger-causality to identify the causal relationships, will add value to the existing literature.

### The flow of the analysis

As the initial step to the analysis, descriptive analysis, correlation analysis and IRF graphs were conducted. Granger-causality analysis was tested as the second stage of the analysis. Before accomplishing the Granger-causality test, this study conducted the first- and second-generation unit root tests. Afterwards lag selection criteria was carried out. Then, stability condition was tested, and finally Granger-causality test was performed.

## Empirical results and discussion

This section presents a descriptive analysis of variables under consideration, REC, NREC, $CO_2$ emissions and GDP, followed by a detailed analysis and discussion of Granger-causality test results.

### Descriptive analysis

Table 2 provides descriptive statistics at the global level as well as at the country sub-category level. There are 4560 total observations in the full dataset, of which 870, 2160, 390, and 1140 are for developed, developing, transitional economies and LDCs, respectively. Descriptive statistics show that the highest average GDP growth as 4.02% and highest average REC as 73% in LDCs. Furthermore, developed countries shows the highest average $CO_2$ emissions indicating that developed countries contribute more proportion of $CO_2$ to the world.

Fig 2 presents for boxplots for all variables under consideration by country group. As can be seen, the skewness of the distributions varies depending on the variable and country category. GDP has a fairly normal distribution for all country groups under consideration while REC demonstrates a positively skewed distribution, except in the LDCs, which shows a negatively skewed distribution. The distribution of NREC is the exact opposite of that of the REC, where all country groups exhibit a negatively skewed distribution except LDCs show a positive skewedness. $CO_2$ indicates an approximately positive skewness across all country groups. While the developed and developing countries have higher carbon emission levels, the LDCs have the lowest range of carbon emission levels.

The Violin plots shown in Fig 3 indicate that the GDP of developed countries is clustered together, suggesting a lower variance among the data points. Furthermore, it suggests that the data points of the GDP of developed countries are distributed around the mean. Broadly, a

**Table 2. Descriptive statistics of country categories.**

| | | GDP | REC | NREC | CO$_2$ |
|---|---|---|---|---|---|
| All Countries | Obs | 4,560 | 4,560 | 4,560 | 4,560 |
| | Mean | 3.5269 | 35.7172 | 64.2827 | 3.7239 |
| | SD | 5.9021 | 31.08463 | 31.0846 | 4.5574 |
| | MIN | -64.0471 | 0.0017 | 1.6600 | 0.0009 |
| | MAX | 149.9730 | 98.3400 | 99.9982 | 31.7785 |
| Developed Countries | Obs | 870 | 870 | 870 | 870 |
| | Mean | 2.2892 | 14.6202 | 85.3797 | 9.0366 |
| | SD | 2.9361 | 13.4035 | 13.4035 | 4.4169 |
| | MIN | -14.1153 | 0.3348 | 37.6300 | 3.4050 |
| | MAX | 25.17624 | 62.3700 | 99.6651 | 30.3617 |
| Developing Countries | Obs | 2,160 | 2,160 | 2,160 | 2,160 |
| | Mean | 3.9831 | 28.0851 | 71.9148 | 3.1292 |
| | SD | 6.2456 | 24.4948 | 24.4948 | 3.8439 |
| | MIN | -64.0471 | 0.00903 | 7.7369 | 0.1431 |
| | MAX | 149.9730 | 92.2630 | 99.9909 | 31.7785 |
| Economies in Transition | Obs | 390 | 390 | 390 | 390 |
| | Mean | 2.2931 | 15.1534 | 84.8465 | 5.1344 |
| | SD | 8.9556 | 17.9644 | 17.9644 | 4.0446 |
| | MIN | -44.9000 | 0.0017 | 35.4200 | 0.3254 |
| | MAX | 35.3845 | 64.5800 | 99.9982 | 15.4412 |
| Least-developed Countries | Obs | 1,140 | 1,140 | 1,140 | 1,140 |
| | Mean | 4.0291 | 73.3135 | 26.6864 | 0.3138 |
| | SD | 5.4025 | 21.7491 | 21.7491 | 0.3224 |
| | MIN | -50.2480 | 0.7000 | 1.6600 | 0.0009 |
| | MAX | 35.2240 | 98.3400 | 99.3000 | 2.7416 |

Note: Obs., SD, MIN, and MAX define Observations, Standard Deviation, Minimum and Maximum, respectively.

Source: Authors' Compilation.

similar pattern could be observed in developing countries, transitional economies and LDCs although the violins demonstrate extreme values in the developing country group.

When considering the REC, apart from the LDCs, most of the data points in REC are grouped below the mean in all country groups. The median REC of LDCs is greater than that of other countries. We also observe that distribution variance in REC across developing countries is higher than that of other country groups. Due to the perfect correlation between REC and NREC, the NREC displays exactly the opposite of the findings of REC. This observation indicates that while the developed countries use more non-renewable energy sources, the LDCs are predominantly relying on renewable energy sources.

With regard to CO$_2$ emissions, the violin plot in figure indicates that the developed countries contribute the most to global carbon emissions and has the highest media value. However, the CO$_2$ emissions data points have a lower variance in developed countries since they are almost uniformly clustered within the distribution. On the other hand, LDCs have the lowest carbon contribution. In summary, the violin plots reveal that the LDCs use more renewable energy sources and emit less CO$_2$ than other economies while the NREC use and CO$_2$ emissions transitional economies and developed countries higher than those of developing countries.

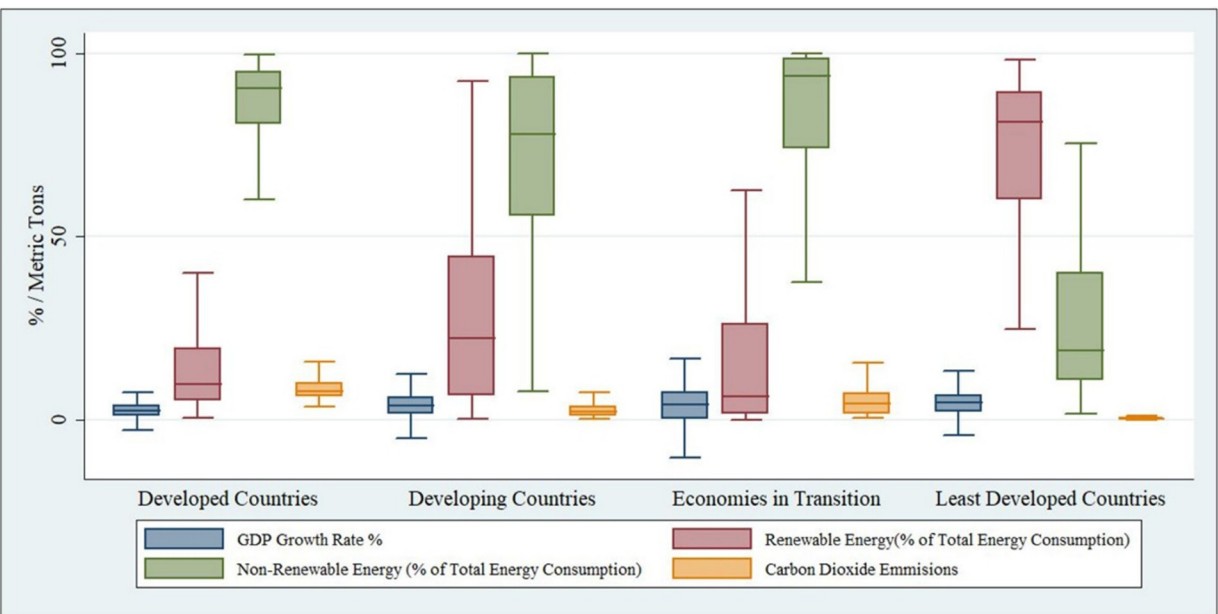

**Fig 2. Box plots of economic growth, energy consumption and CO$_2$ emission by country group.** Source: Authors' Compilation.

Table 3 presents the corelation matrix with respect to the variables under consideration in this study. All estimated correlation coefficients and corresponding standard error estimates revealed a presence of statically significant (at 1% level of significance) linear relationship between the variables. The estimated coefficients show a weak positive (0.082) relationship between GDP and REC, which supports the results of weak negative (-0.082) relationship between GDP and NREC [15, 30]. This could be as a result of the perfect correlation between

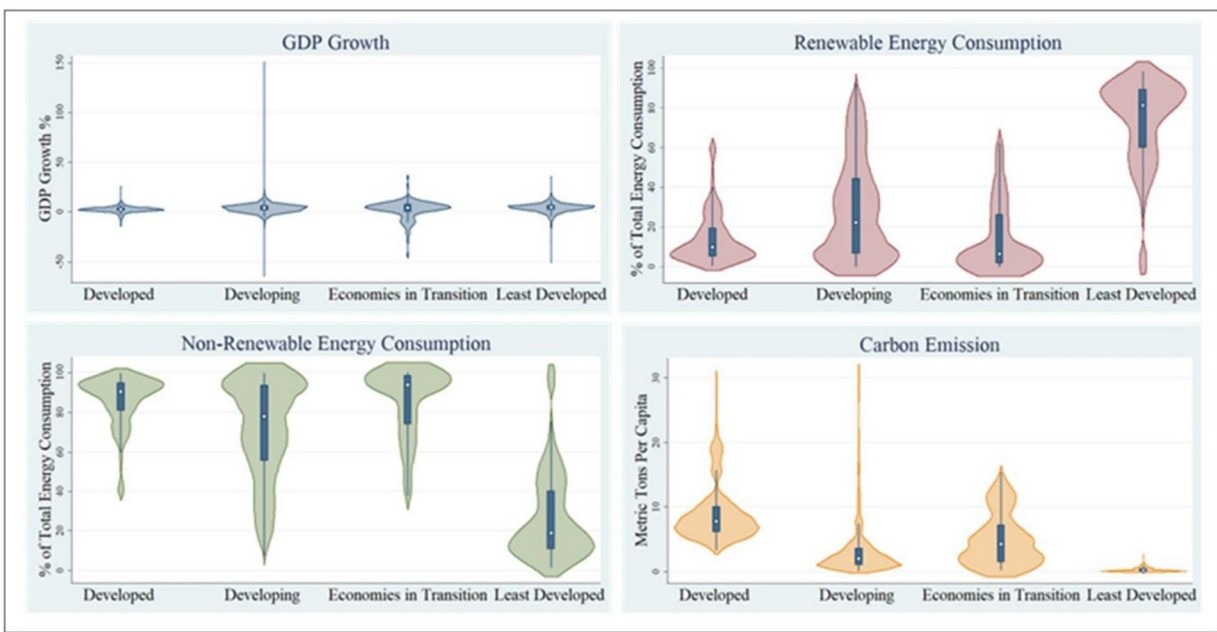

**Fig 3. Violin plots of economic growth, energy consumption and CO$_2$ emission by country group.** Source: Authors' Composition.

**Table 3. Correlation matrix by country group.**

| | GDP | REC | NREC | $CO_2$ |
|---|---|---|---|---|
| **All Countries** | | | | |
| **GDP** | 1.0000 | - | - | - |
| **REC** | 0.0829*** | 1.0000 | - | - |
| **NREC** | -0.0829*** | -1.0000*** | 1.0000 | - |
| **$CO_2$** | -0.0585*** | -0.5930*** | 0.5930*** | 1.0000 |
| **Developed Countries** | | | | |
| **GDP** | 1.0000 | - | - | - |
| **REC** | -0.0582*** | 1.0000 | - | - |
| **NREC** | 0.0582*** | -1.0000*** | 1.0000 | - |
| **$CO_2$** | 0.1203*** | -0.3236*** | 0.3236*** | 1.0000 |
| **Developing Countries** | | | | |
| **GDP** | 1.0000 | - | - | - |
| **REC** | 0.0342*** | 1.0000 | - | - |
| **NREC** | -0.0342*** | -1.0000*** | 1.0000 | - |
| **$CO_2$** | 0.0289*** | -0.4815*** | 0.4815*** | 1.0000 |
| **Economies in Transition** | | | | |
| **GDP** | 1.0000 | - | - | - |
| **REC** | 0.0512*** | 1.0000 | - | - |
| **NREC** | -0.0512*** | -1.0000*** | 1.0000 | - |
| **$CO_2$** | -0.0767*** | -0.6704*** | 0.6704*** | 1.0000 |
| **Least-developed Countries** | | | | |
| **GDP** | 1.0000 | - | - | - |
| **REC** | 0.0762*** | 1.0000 | - | - |
| **NREC** | -0.0762*** | -1.0000*** | 1.0000 | - |
| **$CO_2$** | -0.0066*** | -0.5441*** | 0.5441*** | 1.0000 |

Note: *** Significant at 1% level.

Source: Authors' Compilation.

REC and NREC data. However, these results differ from [58] in the BRICST, which demonstrated the positive effect of economic growth on renewable energy usage. A weak negative (-0.058) relationship was observed between GDP and $CO_2$. The correlation coefficient of -0.59 shows a moderately negative relationship between REC and $CO_2$ while NREC and $CO_2$, on the other hand, shows a moderately positive relationship with a correlation coefficient of 0.59. GDP and $CO_2$ indicate a negative linear relationship in developed and developing countries in transitional economies and LDCs whereas in developed and developing countries, a positive linear relationship exist. Overall, the correlation is weak. However, our results are different to the study which found that economic growth reduce $CO_2$ emissions in both short and long-term in Nordic countries [36]. Furthermore, another study found that long-run dynamic connections exist among industrial $CO_2$ emissions and oil price [85]. There is a positive linear relationship between NREC and $CO_2$ in all country groups, the relationship in the developed country group is somewhat weaker compared to other country groups. In general, the matrix shows a moderate relationship between NREC and $CO_2$ emissions in all countries, but the strongest (0.6704) is seen in transitional economies, suggesting a stronger correlation between NREC and $CO_2$ emissions in transition economies than in other countries. S2 Appendix shows the scatter plots corresponding to Table 3.

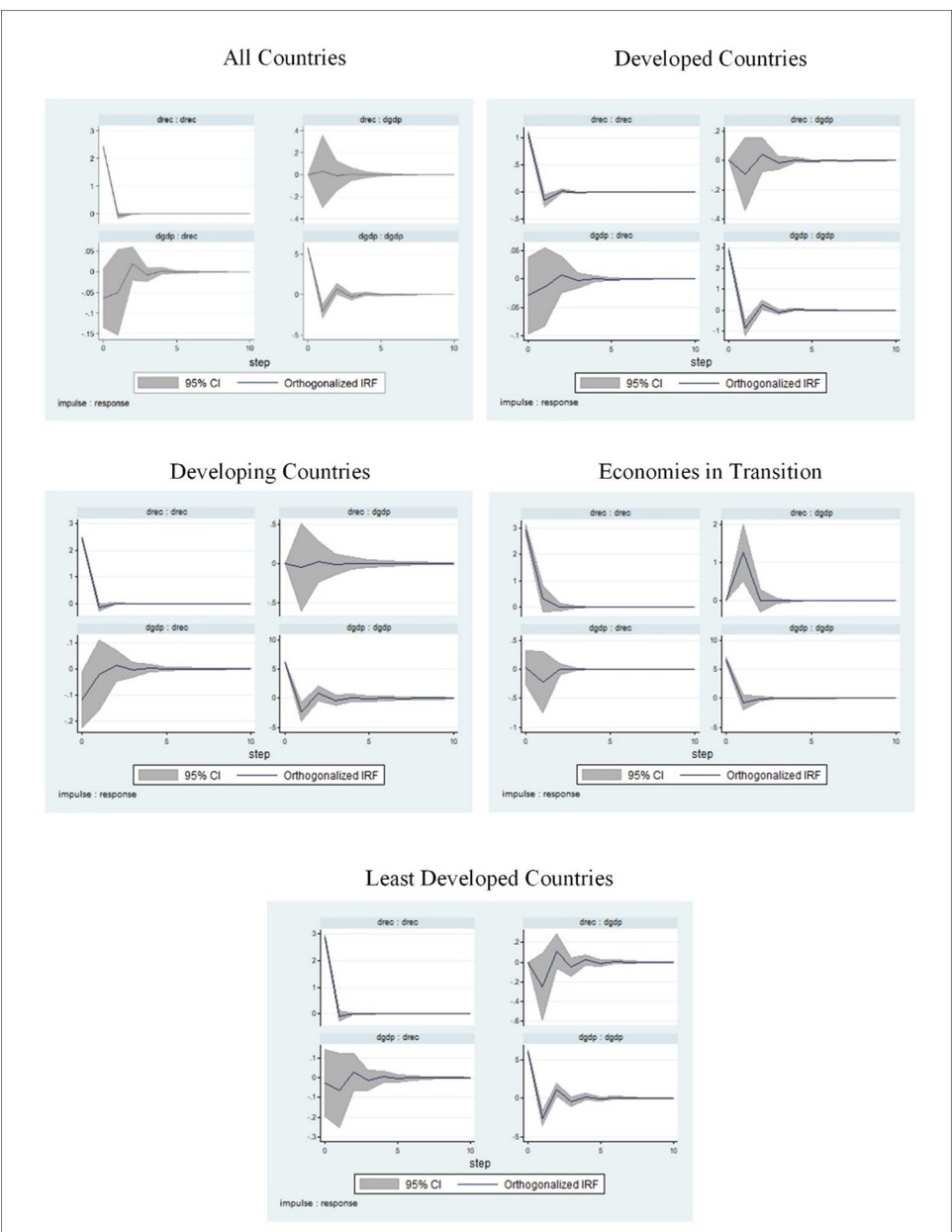

**Fig 4. IRF plots for GDP-REC.** Source: Authors' Compilation.

The IRF plot for GDP and REC in Fig 4 shows that a positive shock to DGDP causes DREC in terms of a 10 year period. The DGDP impulse will result in higher DREC in the first and second periods. Still, it will decrease DREC in the third period, according to the variable IRF analysis. However, the DREC will remain steady and smooth over the long-run globally. In developed countries, the DREC will rise, peaking around the second period due to the DGDP impulses before slowly decreases and becomes smooth after the fifth period. In developing countries, there is a slight increase in REC, and after the fifth period, it becomes smooth. There is a decrease in REC in the first period in transitional economies, and in the second period it will increase, and then stabilises in the long-run. Although there are significant

fluctuations in REC in the LDCs, in the long-run it stabilises. Our findings are however slightly different IRF plot results of the [86], which found that shocks to renewable energy limit economic growth. This is because renewable energy has not reached a particular level to contribute sub-Saharan Africa's economic growth significantly. On the other hand, in a global study of 106 countries classified by different income groups noticed that innovations in most types of energy consumption positively impact economic development [87].

The IRF plots on GDP and NREC given in Fig 5 suggest a negative shock from DGDP to DNREC. Furthermore, at the global level, the DGDP impulse will decrease DNREC in the first and second periods, followed by a sharp increase in DNREC in the third period. The DNREC will remain steady and uniform over the long-run globally. In developed countries, the DNREC will progressively rise and smooth out due to the DGDP impulses after the fourth period. DNREC decreases gradually in develop countries and stabilises after the fifth period. In transitional economies, the NREC increases during the first period of and then decreases and becomes smooth in the long-run. NREC in the LDCs fluctuates substantially until the fifth period and stabilises in the long-run.

The IRF graph for GDP and $CO_2$ shown in Fig 6 demonstrates substantial fluctuations in the impact of DGDP on $DCO_2$ up to period 5 and stabilises thereafter. The impulses of DGDP in developed nations will result in significantly reduced $DCO_2$ in period one, which will then gradually increase in the second period before it slightly decreases again in the third time period. After the fourth period, impulses of DGDP on $DCO_2$ appear to stabilise. Nevertheless, our findings are unique and somewhat different to IRF results [87], which proved that GDP positively affect $CO_2$ emissions. Relatedly, another study show that a shock to GDP results in a reduction of $CO_2$ emissions in sub-samples; British ex-colonies, French ex-colonies and the rest of Stabilization and Association Agreement countries [86]. Overall, the IRF results indicate that additional policy measures are required to reduce $CO_2$ emissions while improving economic growth and REC.

### Granger-causality analysis

Before we carried out the Granger-causality test, we examined whether the data series are station, using the LLC unit root tests. All the variables are significant at the level of 1% for each country category as well as in the global level and the data set is stationary. The test results presented in Table 4 suggest that in all variables, the null hypothesis of series contains unit roots could be rejected at the 1% level of significance, in favour of the alternative hypothesis that the series is stationary.

Furthermore, second generation unit root test introduced by [88] was conducted in advance. The result for the test is shown in S3 Appendix and the final conclusion of the results are the same as in the LLC unit root test.

To determine the causal relationship between GDP, REC, NREC, and $CO_2$ emissions, number of lags for each variable was defined at the lag selection stage. The minimum value that the lags included served as the foundation for the lag selection process. The Minimum Biofilm Inhibitory Concentration test was chosen based on a minimum value. Lag 1 was chosen for the rest of the analysis.

Panel vector autoregression model estimates are typically difficult to interpret on their own [89]. Researchers are frequently interested in how each endogenous variable in the pVAR system will alter in response to external changes [90, 91]. Therefore, IRFs must be carried out first to ensure that the estimated pVAR is stable. For each country category, the corresponding eigenvalue graph, shown in S4 Appendix, shows that all variables satisfy the stability condition.

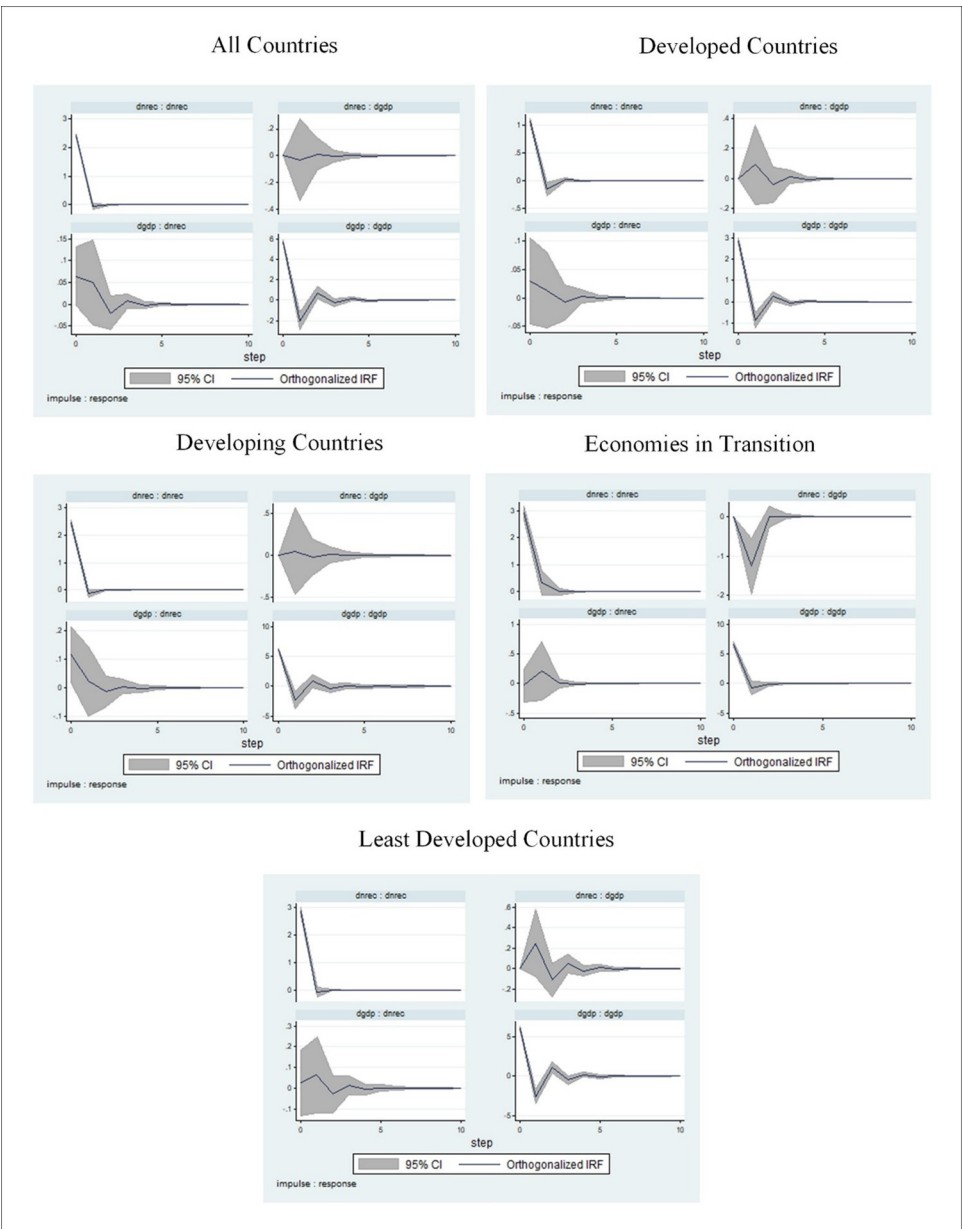

**Fig 5. IRF plots for GDP-REC.** Source: Authors' Compilation.

It is also evident from the graph that all the eigenvalues fall inside the unit circle and that pVAR satisfies the stability condition.

After checking whether the panels contain unit roots, choosing the lag lengths, and testing the stability condition, the Ganger-causality test was carried out to determine the causality between GDP to REC, GDP to NREC, and GDP to $CO_2$ under both directions. The results of the Granger-causality test is presented in Tables 5 and 6. Table 5 presents the Granger-causality test results for GDP and REC, and GDP and NREC. As can be seen, a uni-directional causality running from GDP to REC and GDP to NREC was observed in transitional economies. This observation, however, is considerably different to the findings [11], which revealed a statistically insignificant causal relationship between the REC and economic growth in Russia; a

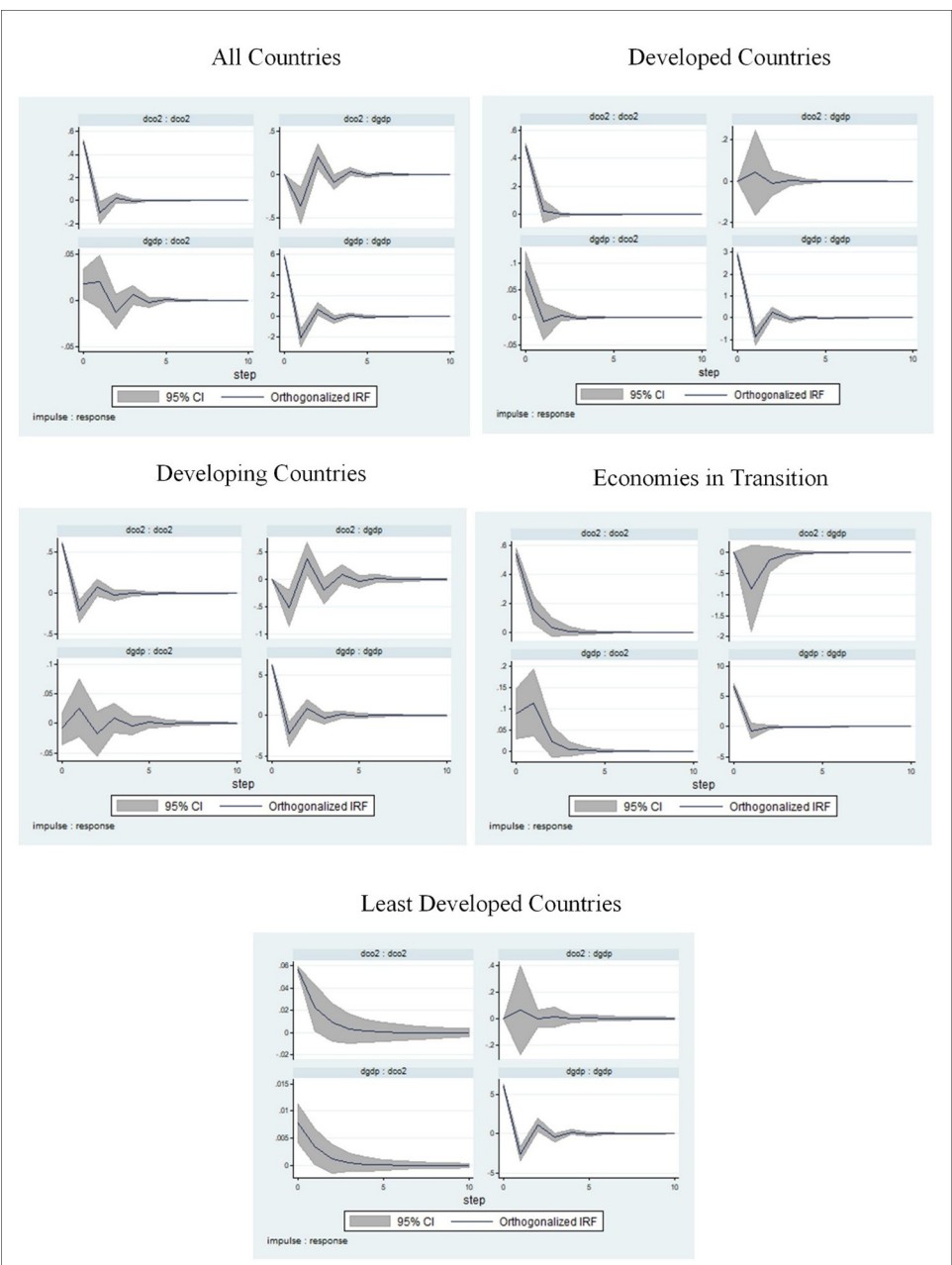

**Fig 6. IRF plots for GDP-$CO_2$.** Source: Authors' Compilation.

transitional economy. No other significant causal relationship was found between GDP and REC or GDP and NREC in any other country group in our study. The study based on the BRICS region also revealed that the NREC, and economic growth had a bi-directional causative link with the environmental impact, but not with REC [50]. Hence, our results support the existing literature which also conclude that there is no causal relationship between GDP and REC [40, 65]. However, our results are somewhat different to the existing literature on USA [11], Pakistan [14], Italy [27] and developing countries [53]. Similarly, our results are also inconsistent with a study [25], which disclosed a uni-directional causal link between economic

**Table 4. LLC unit root test.**

|  | DGDP | DREC | DNREC | DCO$_2$ |
|---|---|---|---|---|
| **All Countries** | -46.1213*** | -24.6919*** | -24.6921*** | -24.4088*** |
| **Developed Countries** | -19.6194*** | -8.5691*** | -8.5692*** | -9.1687*** |
| **Developing Countries** | -31.7354*** | -15.3214*** | -15.3216*** | -16.2570*** |
| **Economies in Transition** | -14.3468*** | -12.2068*** | -12.2070*** | -10.1444*** |
| **Least-developed Countries** | -23.6279*** | -13.0417*** | -13.0417*** | -12.7696*** |

Note: LLC unit root test—H$_0$: series contains unit roots and H$_1$ series is stationary.

*** Significant at 1% level.

Source: Authors' Compilation.

growth and energy utilize in the top 5 carbon emitting countries (China, the United States, India, Brazil, and Russia). Furthermore, our results which found a bi-directional causal relationship between GDP and NREC in the African OPEC region, are also substantially different to the literature [49]. And also results of another study has been revealed a bi-directional relationship between GDP and REC in emerging and developing countries and selected LDCs [17, 49, 92]. The differences in findings in the literature could be attributed to various factors, such as differences in econometric techniques used and time period used in the analysis. Therefore, it is crucial to consider social and environmental elements while making judgments on REC and NREC. S5 Appendix demonstrates the Granger-causality results on REC, NREC and GDP for each country category by using the field maps.

**Table 5. Summary of Granger-causality test for REC, NREC and gross domestic production.**

| Renewable Energy Consumption | | | |
|---|---|---|---|
|  | DGDP ↓ DREC | DREC ↓ DGDP | DGDP—DREC |
| **All Countries** | 0.0320 | 1.1380 | ⬌ |
| **Developed Countries** | 0.5000 | 0.2650 | ⬌ |
| **Developing Countries** | 0.0340 | 0.2340 | ⬌ |
| **Economies in Transition** | 11.6320*** | 0.7530 | → |
| **Least-developed Countries** | 2.0990 | 0.4900 | ⬌ |
| **Non-renewable Energy Consumption** | | | |
|  | DGDP ↓ DNREC | DNREC ↓ DGDP | DGDP—DNREC |
| **All Countries** | 0.0320 | 1.1380 | ⬌ |
| **Developed Countries** | 0.5000 | 0.2650 | ⬌ |
| **Developing Countries** | 0.0340 | 0.2340 | ⬌ |
| **Economies in Transition** | 11.6320*** | 0.7530 | → |
| **Least-developed Countries** | 2.0990 | 0.4900 | ⬌ |

Note: H$_0$ is x(t) does not Granger-cause y(t) and H$_1$ is x(t) does not Granger-cause y(t) The characters ⬌ and → represents a bi-directional, no causal relationship, one-way-left direction and one-way-right direction causal relationship, respectively.

* Significant at 10% level,

** Significant at 5% level, and

*** Significant at 1% level.

Source: Authors' Composition.

**Table 6. Granger-causality test results for $CO_2$ emissions and GDP.**

| | DGDP ↓ DCO₂ | DCO₂ ↓ DGDP | DGDP—DCO₂ |
|---|---|---|---|
| **All Countries** | 12.2010*** | 2.3890 | → |
| **Developed Countries** | 0.1630 | 0.5680 | ⊕ |
| **Developing Countries** | 10.7800*** | 0.7930 | → |
| **Economies in Transition** | 2.9720* | 4.8170** | ↔ |
| **Least-developed Countries** | 0.1260 | 0.0570 | ⊕ |

Note: $H_0$ is x(t) does not Granger-cause y(t) and $H_1$ is x(t) does not Granger-cause y(t) The characters ↔ ⊕ and →
represents a bi-directional, no causal relationship, and one-way-right direction causal relationship, respectively.
* Significant at 10% level,
** Significant at 5% level, and
*** Significant at 1% level.
Source: Authors' Composition.

The findings of the Granger-causality test for GDP to $CO_2$ and $CO_2$ to GDP are presented in Table 6. As can be seen, GDP Granger causes $CO_2$ in all countries and developing countries. However, this observation is somewhat different from certain studies. For instance, a study conducted in Thailand found no significant causal relationship between GDP and $CO_2$ [2]. In transitional economies, there is a bi-directional causality between GDP and $CO_2$. However, no causal relationship between GDP and $CO_2$ was found in developed countries or LDCs. This observation, however, contradicts with the findings of a study [49], that there is a bi-directional causal relationship between economic growth and $CO_2$ emissions in some LDCs in African region; Angola, Niger. Similarly, our results are also differ from a study [42], that identified a uni-directional causal relationship running from GDP to $CO_2$ in 22 developed countries. Similarly, a study [24] identified a uni-directional causal interaction from economic growth to $CO_2$ emissions in the top 10 energy transition economies (China, USA, UK, Germany, France, Japan, India, Spain, Brazil, and South Korea). The Granger-causality results on $CO_2$ and GDP for each country category are illustrated in S6 Appendix.

Furthermore, in line with a previous study [93], lag-based sensitivity analysis was conducted to explore whether results are insensitive to changes in the model. Results for the lag-based analysis presented in S7 Appendix. However, according to the lag selection process for the current study, lag 1 is the most suitable for the analysis and the analysis was conducted thereon.

## Concluding comments

This study examined the relationship between renewable and non-renewable energy, $CO_2$ emissions, and economic growth. The data for 152 countries for the period 1990–2019 was sourced from the World Bank open database. Although there is a significant body of literature on the inter-relationships between the variables under consideration in this study, there are no consensus in the findings. By focusing on the nexus between these variables at the global level as well as at different country group level (developed and developing countries, transitional economies and LDCs) this study further expands the existing body of literature in general while contributing to the limited literature in the context of transitional economies and LDCs in particular. We utilised various analytical techniques, such as box plots, violin plots, IRF plots, correlation analysis and Granger casualty tests to investigate the link between these variables.

Our analysis indicates that developed nations consume a higher proportion of non-renewable energy in their overall energy consumption. This may have a direct impact on the $CO_2$

emissions by developed countries. In fact, the descriptive analysis in this study revealed that the $CO_2$ emissions of developed countries are substantially higher than those of other nations. Furthermore, the descriptive analysis of this study revealed that the LDCs emit less $CO_2$ than other nations and use more renewable energy to fulfil their energy needs. In particular, the scatterplots which depict a negative linear relationship between REC and $CO_2$ and a positive linear relationship NREC and $CO_2$, supports this proposition. Therefore, our findings further support global initiatives for adopting renewable energy as their primary energy source and impose appropriate rules and processes in place to limit carbon emissions mitigate environmental issues worldwide.

The results of the Granger-causality test show that outside of transitional economies, there is no Granger-causal relationship between GDP and REC or GDP and NREC. In transitional economies, there exists a uni-directional causality running from GDP to REC and GDP to NREC. This suggest that economic growth in these countries drives the REC and NREC, supporting the well-known conservation hypothesis [45]. The presence of the conservation hypothesis indicate that energy conservation policies could be implemented with limited or no adverse impact on the progress of the economic activities. The presence of uni-directional relationship between GDP and energy consumption in transitional economies also suggest that making decisions involving both variables, policymakers should consider the pertinent impacts and undertake a sensitivity analysis because both factors impact each another.

Our results also found that there is a uni-directional causality running from GDP to $CO_2$ for all countries (mainly driven by such relationship in developing countries). No Granger-causality has been observed in developed or LDCs. Furthermore, results suggest that there is a bi-directional causality between GDP and $CO_2$ in transitional economies. In summary, despite the lack of evidence of a causal relationship between GDP and REC or GDP and NREC, this study uncovers some useful insights on transitional economies—the presence of a bi-directional causal relationship between GDP and $CO_2$. Given the result that GDP granger-causes $CO_2$ emissions, developing economies, developed economies as well as transitional economies need take serious actions to reduce $CO_2$ emissions.

The findings of this study are not without limitations. This research does not focus on the social impacts of employing renewable energy sources, including job creation, improved rural life quality, improved public health by lowering pollutants, and increased awareness among professionals and the public. Future scholars might reassess the economic impacts of renewable energy by considering several drivers and elements, including foreign direct investment, public and private institutions, technical innovation, R&D, corruption, and the stability of the nation's finances.

## Policy implications

The findings of this study provide useful policy insights in the areas of REC, NREC, GDP growth and $CO_2$ emissions, which are summarised below.

Based on this analysis, the main recommendation is that all countries, particularly the developed countries need to take necessary initiative to reduce overall $CO_2$ levels to achieve sustainable economic growth. Majority of the developed countries have higher GDP levels and production capacities which may lead to increased energy consumption from non-renewable energy sources compared to other nations. Policymakers should focus more on adopting renewable energy sources to produce energy as this may contribute to lower $CO_2$ emissions. All the energy related investments and developments in developed countries should give priority to and include renewable energy as a key performance indicator in their investment appraisal considerations.

Similar concerns are applicable in the context of transitional economies. Nevertheless, policymakers should focus on both GDP and $CO_2$ when making decisions because this study's results indicated a bi-directional causality between the two variables in transitional economies. Hence, policy decisions related to sustainable economic growth in transitional economies should be carefully analysed before implementing due to this causal relationship. Therefore, to mitigate the adverse environmental implications of increased NREC, it is worthwhile for transitional economies to invest more on renewable energy sources and implement strategies to reduce $CO_2$ emissions.

Ultimately, considering the overall benefits, it is advisable for all the nations to go for more renewable energy sources for a sustainable future. $CO_2$ emission is one of the most pressing environmental issues nowadays and increasing cleaner and REC will have a direct considerable impact on reducing $CO_2$ emissions. All the nations should contribute to the world economy by setting up clear pathways to achieve sustainable economic goals. In general, the findings of this current study imply that it is preferable to reduce the $CO_2$ emissions by implementing necessary policies while improving GDP in all countries across the globe.

## Supporting information

**S1 Appendix. Data file.**
(XLSX)

**S2 Appendix. Scatter plots of country categories.**
(DOCX)

**S3 Appendix. Second generation unit root test.**
(DOCX)

**S4 Appendix. Graph of eigenvalue stability condition.**
(DOCX)

**S5 Appendix. Filled maps of Granger-causality test for REC, NREC and gross domestic production.**
(DOCX)

**S6 Appendix. Filled maps of Granger-causality test for $CO_2$ and gross domestic production.**
(DOCX)

**S7 Appendix. Sensitivity analysis for LLC unit root test & Granger-causality test.**
(DOCX)

## Author Contributions

**Conceptualization:** Hasara Dissanayake, Nishitha Perera, Sajani Abeykoon, Diruni Samson, Ruwan Jayathilaka.

**Data curation:** Hasara Dissanayake, Nishitha Perera.

**Formal analysis:** Hasara Dissanayake, Nishitha Perera, Ruwan Jayathilaka.

**Investigation:** Sajani Abeykoon, Diruni Samson, Ruwan Jayathilaka.

**Methodology:** Hasara Dissanayake, Nishitha Perera, Ruwan Jayathilaka.

**Project administration:** Ruwan Jayathilaka.

**Resources:** Diruni Samson.

**Software:** Hasara Dissanayake, Nishitha Perera.

**Supervision:** Ruwan Jayathilaka, Maneka Jayasinghe, Shanta Yapa.

**Validation:** Hasara Dissanayake, Sajani Abeykoon.

**Visualization:** Hasara Dissanayake, Nishitha Perera.

**Writing – original draft:** Hasara Dissanayake, Nishitha Perera, Sajani Abeykoon, Diruni Samson, Ruwan Jayathilaka, Maneka Jayasinghe.

**Writing – review & editing:** Ruwan Jayathilaka, Maneka Jayasinghe, Shanta Yapa.

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
