## [Decision Letter · Decision Letter 0]

18 Apr 2023

PONE-D-23-08466Nexus between Carbon Emissions, Energy Consumption, and Economic Growth: Evidence from Global Economies

PLOS ONE

Dear Dr. Jayathilaka,

Thank you for submitting your manuscript to PLOS ONE. After careful consideration, we feel that it has merit but does not fully meet PLOS ONE’s publication criteria as it currently stands. Therefore, we invite you to submit a revised version of the manuscript that addresses the points raised during the review process.

We look forward to receiving your revised manuscript.

Kind regards,

Roni Bhowmik, Ph.D.

Academic Editor

PLOS ONE

Journal Requirements:

Additional Editor Comments:

I believe that the authors will carefully address all of reviewers concern and have manage to further improve the manuscript’s quality. 

Authors can also use the following points below as a guideline to help them come out with an interesting that is more scientific.

1. Background & Significance: (What general background does the reader need to understand the manuscript and how important is it in scientific research).

Add a few more sentences at the very beginning of the introduction explaining your paper’s contribution or attempts for dealing with or presenting solutions for a specific problem/s and your special contribution to this research paper.

Motivations & Objectives: (Why are you conducting the study and what are the goals to achieve?)

2. Please present your literature review in a concise SmartArt chart format.

3. Policy implications are not strongly related to the research findings, so it needs to be improved.

Reviewers' comments:

Reviewer's Responses to Questions

**Comments to the Author**

1. Is the manuscript technically sound, and do the data support the conclusions?

Reviewer #1: Yes

Reviewer #2: Partly

Reviewer #3: Partly

Reviewer #4: Yes

2. Has the statistical analysis been performed appropriately and rigorously? 

Reviewer #1: Yes

Reviewer #2: No

Reviewer #3: Yes

Reviewer #4: N/A

3. Have the authors made all data underlying the findings in their manuscript fully available?

Reviewer #1: Yes

Reviewer #2: Yes

Reviewer #3: Yes

Reviewer #4: No

4. Is the manuscript presented in an intelligible fashion and written in standard English?

Reviewer #1: Yes

Reviewer #2: Yes

Reviewer #3: Yes

Reviewer #4: Yes

5. Review Comments to the Author

Reviewer #1: The manuscript titled “Nexus between Carbon Emissions, Energy Consumption, and Economic Growth: Evidence from Global Economies” has been reviewed. Here are my minor suggestions which can improve the manuscript.

First of all, the idea of the analysis is very relevant. In fact, given the urgency of climate change, all research focused on the economic events that could harm the environment are vital. Briefly, I have a good time reading this paper, but some revisions are needed before being published.

Here are my comments:

• Overall, I recommend the inclusion of an acronyms table before the introduction section. In my opinion, there is a lot of acronyms, which could make it difficult to the readers to have it present along all paper. So, a table might help.

• The introduction part is required to add few more sentences to increase the strength of this article and kindly bring in the research problem, objective, novelty and explain it in last paragraph of the section of Introduction. Information from these studies is need in the introduction section

https://doi.org/10.1016/j.jenvman.2022.116884

https://doi.org/10.1007/s10668-023-03085-4

• Where is the significance of study of this research? Its missing. Kindly incorporate it

such information since it is relevant to your study

• The literature review section is weak. Kindly incorporate the following recent studies

https://doi.org/10.1002/gj.4664

https://doi.org/10.1080/13504509.2022.2162147

https://doi.org/10.1007/s11356-022-22785-4

https://doi.org/10.1002/gj.4694

• At the end of the literature review, kindly incorporate the gap in the literature

Where is the theoretical Framework? Kindly incorporate it.

• The author(s) need to present the flow of the analysis. The author(s) can incorporate this in the data and method section.

• The discussion section needs minor improvement. This can be done by comparing your results with prior studies. Kindly compare your study with the following studies

https://doi.org/10.1080/13504509.2022.2115577

https://doi.org/10.1177/0958305X231167463

• Please insert graphical presentations for your results.

• Use of English language need slight improvement. Revise almost all sentences in the manuscript with appropriate use of grammar, punctuation and speech.

Reviewer #2: The study aims to examine the relationship between renewable and nonrenewable energy use, CO2 emissions and economic growth in developed, developing, and Least-developed Economies in Transition from 1990 to 2019.

The manuscript suggests that the study included 152 countries and used Granger-causality as a methodology to investigate the links between the variables. The findings of the study indicated that there may or may not be a relationship between the consumption of renewable and non-renewable energy sources and economic growth.

Furthermore, there was no Granger-causal relationship between GDP and renewable energy consumption outside of Economies in Transition, and the GDP and CO2 of all countries had a one-way relationship.

While the manuscript is organized and well-written with chapters that follow a scientific study, the methodology used in the study could be improved. The reviewer suggests that the authors use additional econometrics tests to support the discussions and conclusions. The study's great literature review and inclusion of sufficient studies on the topic are positives. However, the findings do not contribute significantly to the literature.

The study highlights that it is crucial to take action to lower CO2 emissions, which would develop renewable energy while also stimulating the economy. While the study provides an analysis of the variables discussed in the four subgroups, there is a need for more empirical studies that encompass all regions.

In conclusion, the manuscript is partly technically sound but needs improvement in methodology, particularly in the use of econometric techniques. I would suggest the used second generation unit-root testing and better causality methodology.

The findings of the study do not contribute significantly to the literature, but the study has a great literature review, and sufficient studies are provided on the topic. The authors are advised to review the econometric techniques used for the study as there are better techniques available.

Reviewer #3: 1- The novelty of paper is extremely weak. The authors are suggested to make the manuscript novel in terms of methodology (atleast). Otherwise, the manuscript seems missfit for publication.

2- The literature needs to be significantly improved. The authors can use these studies to update the literature review section:

Monetary and energy policy interlinkages: The case of renewable energy in the US; Investigating the Impact of Climate Change Mitigation Technology on the Transport Sector CO2 Emissions: Evidence From Panel Quantile Regression; Does sectoral energy consumption depend on trade, monetary, and fiscal policy uncertainty? Policy recommendations using novel bootstrap ARDL approach; Exploring the dynamic relationship between energy efficiency, trade, economic growth, and CO2 emissions: Evidence from novel fourier ARDL approach; Does green environmental policy promote renewable energy consumption in BRICST? Fresh insights from panel quantile regression; Do geopolitical risk and energy consumption contribute to environmental degradation? Evidence from E7 countries;; Applying a dynamic ARDL approach to the Environmental Phillips Curve (EPC) hypothesis amid monetary, fiscal, and trade policy uncertainty in the USA; Investigating the Environmental Kuznets Curve hypothesis amidst geopolitical risk: Global evidence using bootstrap ARDL approach; Do economic policy uncertainty and geopolitical risk surge CO2 emissions? New insights from panel quantile regression approach; Impact of economic policy uncertainty on CO2 emissions in the US: Evidence from bootstrap ARDL approach; Impact of economic policy uncertainty on CO2 emissions: evidence from top ten carbon emitter countries

3- The motivation behind the employed methodology seems weak. The authors sould also work on it.

4- There is a need to perform sensitivity analysis to explore whether results are insensitive to changes in model/methodology/data.

5- Results should be explained in a detailed manner

6- policy implications should be in line with the main findings.

Reviewer #4: Comments to the Author

1. The Abstract must report the aim of the study, the basic information on the sample (time span, countries analyzed), the empirical methodology used, the main findings, and the relevant policy implications.

2. The Introduction should highlight the relevance of the topic, the novelty of the results, the importance of policy implications, the sample’s choice, the methodology’s appropriateness, the data used, the contribution to the literature, and the limitations of the study.

3. The literature review is partial and incomplete, and some recent and relevant contributions should be cited and discussed .i.e., Role of technological innovation and globalization in BRICS economies: policy towards environmental sustainability. International Journal of Sustainable Development & World Ecology, 29(7), 593-610; Can green resource productivity, renewable energy, and economic globalization drive the pursuit of carbon neutrality in the top energy transition economies?. International Journal of Sustainable Development & World Ecology, 1-15; Determinants of carbon emissions in Argentina: The roles of renewable energy consumption and globalization. Energy Reports, 7, 4747-4760; Toward a sustainable environment and economic growth in BRICS economies: do innovation and globalization matter?. Environmental Science and Pollution Research, 29(38), 57740-57757; Does globalization moderate the effect of economic complexity on CO2 emissions? Evidence from the top 10 energy transition economies. Frontiers in Environmental Science, 555; Testing the asymmetric effect of financial stability towards carbon neutrality target: The case of Iceland and global comparison. Gondwana Research.

5. The gap in literature should be reported in the literature section.

6. How the authors have derived the empirical model? There should be a thorough theoretical underpinning behind the model. This section should be based on the logic of the authors, and no citation/reference should appear here. This section will be followed by the empirical model.

7. Conclusion reiterates the results, which is completely undesirable. The authors should summarize the results within a maximum of 3 sentences. Moreover, the policies are completely vague, and it seems that the authors already had them in mind before starting the paper. The policies should be directly derived from the discussion of the results, and they should not go beyond the results.

6. PLOS authors have the option to publish the peer review history of their article (what does this mean?). If published, this will include your full peer review and any attached files.

Reviewer #1: No

Reviewer #2: No

Reviewer #3: No

Reviewer #4: No

---

## [Author Response · Author response to Decision Letter 0]

29 May 2023

Point by point response to editor and reviewers.

We would like to express our profound appreciation to the editor for giving us the opportunity to revise and resubmit the paper. We are also grateful to the reviewers for the constructive comments and suggestions made on our manuscript which we found very helpful in revising and improving our paper. Given below is a detailed description of how we have addressed each comment in the revised version of the paper.

Editor’s comment 1: Background & Significance: (What general background does the reader need to understand the manuscript and how important is it in scientific research).

Add a few more sentences at the very beginning of the introduction explaining your paper’s contribution or attempts for dealing with or presenting solutions for a specific problem/s and your special contribution to this research paper.

Motivations & Objectives: (Why are you conducting the study and what are the goals to achieve?)

Authors’ Response to Editor Comment 1: Thank you for the valuable comment. We have further strengthened the significance of the study as a separate paragraph in lines 140-157 in the revised version of the paper. Also, we have further strengthened the introduction in lines 109-133.

“This study classifies nations as developed, developing, transitional economies, and LDCs using data from the World Bank for the years between 1990 and 2019 for 152 countries to determine the relationship between global GDP growth, energy consumption and carbon footprint. The management and use of renewable energy sources are crucial due to the global nature of the energy challenges. Renewable energy is pure, secure, and unbounded, in contrast to conventional energy. Therefore, it is rapidly acquiring momentum worldwide, and as predicted, will soon displace many traditional energy sources and take the lead concerning the overall proportion of energy consumption.

Developing and LDCs continue to experience rapid, unsustainable economic growth. However, by switching to renewable energy, sustainable growth can also be experienced. Using renewable technologies can benefit local power generation in rural and remote areas. Similarly, lowering carbon footprint is essential for preventing climate change, improving sustainable development, generating financial savings, and upholding international commitments. It is a shared obligation that seeks participation and collaboration from all key actors including individuals, groups, and governments worldwide.”

“To this end, the present research contributes to the knowledge base and endeavours to fill gaps in the existing literature in 4 ways; (1) although economic growth, non-renewable energy, and renewable energy nexus has been well-researched in the literature, very little research has been carried out on LDCs or transitional economies. Since these countries are highly dependent on fossil fuel for economic activities, empirical evidence from these countries will provide important insights for policymakers and our research contributes to this limited body of literature; (2) this study extends the knowledge base on this important research area by using a variety of techniques such as violine graphs, panel Granger-causality test, and the impulse-response functions (IRF) plots which have not widely used in the existing research on this topic. Hence, this study provides a further nuanced understanding on the link between energy consumption, CO2 emissions, and economic growth. To ensure the reliability of the findings, before carrying out the Granger-causality test, the Panel Vector Auto-Regression (pVAR) was checked to see if it is satisfied or failed to satisfy the stability criteria for all variables for all countries under consideration; and (3) the majority of existing studies focus either on single country or a small group of countries. Such research provides limited possibilities for comparison across countries with different stages of economic development. This study carries out a global level analysis as well as a disaggregated analysis by developed, developing, LDCs, and transitional economies, enabling a comparison across countries in different stages of economic development; and (4) current study covers a wide time period (1990-2019) and uses the most recent for the empirical analysis. We believe that a comprehensive study at global level, like ours, would enhance the evidence base on this important area of concern to facilitate decision-making. However, this study does not address the creation of jobs, improved quality of life in rural areas, improved public health as a result of reduced pollution, or increased public and professional awareness; it is because the study focused only on renewable energy, environmental sustainability and economic growth.”

Editor’s comment 2: Please present your literature review in a concise SmartArt chart format.

Authors’ Response to Editor Comment 2: Thank you very much and well noted. Flow diagram for literature review has been added as Figure 1 in the revised paper and mentioned in the main body of the revised manuscripts as quoted below in line 200-214.

“In order to carry out this study, a thorough review of the literature was conducted, focusing solely on the effects of renewable energy, non-renewable energy, and CO2 emissions on economic growth concerning four country groups. The Figure 1 presents the systematic approach adopted for selecting the research papers for the literature review. As elaborated in Figure 1, 152 articles were found using keywords and database searches and 83 publications were removed. The remaining 69 papers were divided into four categories: developed countries, developing countries, transitional economies, and LDCs.”

Editor’s comment 3: Policy implications are not strongly related to the research findings, so it needs to be improved.

Authors’ Response to Editor Comment 3: We have noted your comment with many thanks. To address your concern, we have added a new section on policy implications aligning with the main findings of the study in line 889-921.

“The findings of this study provide useful policy insights in the areas of REC, NREC, GDP growth and CO2 emissions, which are summarised below.

Based on this analysis, the main recommendation is that all countries, particularly the developed countries need to take necessary initiative to reduce overall CO2 levels to achieve sustainable economic growth. Majority of the developed countries have higher GDP levels and production capacities which may lead to increased energy consumption from non-renewable energy sources compared to other nations. Policymakers should focus more on adopting renewable energy sources to produce energy as this may contribute to lower CO2 emissions. All the energy related investments and developments in developed countries should give priority to and include renewable energy as a key performance indicator in their investment appraisal considerations. 

Similar concerns are applicable in the context of transitional economies. Nevertheless, policymakers should focus on both GDP and CO2 when making decisions because this study’s results indicated a bi-directional causality between the two variables in transitional economies. Hence, policy decisions related to sustainable economic growth in transitional economies should be carefully analysed before implementing due to this causal relationship. Therefore, to mitigate the adverse environmental implications of increased NREC, it is worthwhile for transitional economies to invest more on renewable energy sources and implement strategies to reduce CO2 emissions.

Ultimately, considering the overall benefits, it is advisable for all the nations to go for more renewable energy sources for a sustainable future. CO2 emission is one of the most pressing environmental issues nowadays and increasing cleaner and REC will have a direct considerable impact on reducing CO2 emissions. All the nations should contribute to the world economy by setting up clear pathways to achieve sustainable economic goals. In general, the findings of this current study imply that it is preferable to reduce the CO2 emissions by implementing necessary policies while improving GDP in all countries across the globe.”

Reviewer 1 General Comments: The manuscript titled “Nexus between Carbon Emissions, Energy Consumption, and Economic Growth: Evidence from Global Economies” has been reviewed. Here are my minor suggestions which can improve the manuscript. 

First of all, the idea of the analysis is very relevant. In fact, given the urgency of climate change, all research focused on the economic events that could harm the environment are vital. Briefly, I have a good time reading this paper, but some revisions are needed before being published.

Authors’ Response to Reviewer 1 general 1: Thank you very much for your remark. We greatly value your constructive comments to improve our manuscript. We have carefully addressed all of your comments in the revised version of the paper.

Reviewer 1 comment 1: Overall, I recommend the inclusion of an acronyms table before the introduction section. In my opinion, there is a lot of acronyms, which could make it difficult to the readers to have it present along all paper. So, a table might help.

Authors’ Response to Reviewer 1 comment 1: Well noted with many thanks, with related to the comment about the acronym table. As per the PLOS One guidelines it is expected to define the abbreviations/acronyms in first appearance in the text, which we have followed in the revised version of the paper.

https://journals.plos.org/plosone/s/submission-guidelines

Quoted from PLOS One guidelines, “Define abbreviations upon first appearance in the text.”

Reviewer 1 comment 2: The introduction part is required to add few more sentences to increase the strength of this article and kindly bring in the research problem, objective, novelty and explain it in last paragraph of the section of Introduction. Information from these studies is need in the introduction section.

https://doi.org/10.1016/j.jenvman.2022.116884

https://doi.org/10.1007/s10668-023-03085-4

Authors’ Response to Reviewer 1 comment 2: Thank you very much for the valuable comment of yours. We have revised the introduction section accordingly. We have also incorporated the suggested references in the revised version of the paper in lines 78-86.

“…The analysis done by some researchers found that renders space for making policy suggestions as government spending on research and development is essential to achieving the goal of investing in renewable energy by all nations [6]. It is essential to create emission-reducing technologies and implement strict environmental regulations in order to reduce ecological footprint [6]. Similarly, the government should adopt an effective and sustainable approach in ensuring energy consumption optimization and economic growth [7]. Furthermore, it has been found that energy consumption results in greater emissions in both short and long term globally….”

Reviewer 1 comment 3: Where is the significance of study of this research? Its missing. Kindly incorporate it

such information since it is relevant to your study

Authors’ Response to Reviewer 1 comment 3: Thank you for the insightful comment. To address your concern, we have added a new section on the significant of the study after the introduction section in lines 140-157 in the revised version of the manuscript. 

“This study classifies nations as developed, developing, transitional economies, and LDCs using data from the World Bank for the years between 1990 and 2019 for 152 countries to determine the relationship between global GDP growth, energy consumption and carbon footprint. The management and use of renewable energy sources are crucial due to the global nature of the energy challenges. Renewable energy is pure, secure, and unbounded, in contrast to conventional energy. Therefore, it is rapidly acquiring momentum worldwide, and as predicted, will soon displace many traditional energy sources and take the lead concerning the overall proportion of energy consumption.

Developing and LDCs continue to experience rapid, unsustainable economic growth. However, by switching to renewable energy, sustainable growth can also be experienced. Using renewable technologies can benefit local power generation in rural and remote areas. Similarly, lowering carbon footprint is essential for preventing climate change, improving sustainable development, generating financial savings, and upholding international commitments. It is a shared obligation that seeks participation and collaboration from all key actors including individuals, groups, and governments worldwide.”

Reviewer 1 comment 4: The literature review section is weak. Kindly incorporate the following recent studies

https://doi.org/10.1002/gj.4664

https://doi.org/10.1080/13504509.2022.2162147

https://doi.org/10.1007/s11356-022-22785-4

https://doi.org/10.1002/gj.4694

Authors’ Response to Reviewer 1 comment 4: Many thanks for this suggestion. The recommended research articles have been incorporated in the literature review of the revised manuscript in lines 324-326, 344-350, 362-366, and 366-369.

“…Some studies' primary suggestions include decreasing fossil fuel subsidies and increasing investment in renewable energy [47]…”

“….On the one hand, a study conducted for the E7 countries (Brazil, China, India, Indonesia, Mexico, Russia, and Turkey) between 1990 and 2015 proved that using renewable energy improves environmental quality by reducing both ecological footprint and CO2 emissions. On the other hand, using non-renewable energy increases both the environmental impact and CO2 emissions. The results of the causality test suggest that there is a uni-directional causal relationship running from NREC to economic growth [52]…”

“…Similarly, in China, CO2 emissions are positively influenced by economic growth, coal consumption, and natural resources at all frequencies, or both in the short and long-run, leading to environmental degradation, using the Wavelet Local Multiple Correlation bivariate cases [56]...”

“….According to the study that utilised quantile-on-quantile regression (QQR) method, Nigeria and Mexico experienced positive effects of energy resources on economic growth in the majority of quantiles [57].…”

Reviewer 1 comment 5: At the end of the literature review, kindly incorporate the gap in the literature.

Where is the theoretical Framework? Kindly incorporate it.

Authors’ Response to Reviewer 1 comment 5: Well noted the mentioned comment with many thanks. As suggested, the research gap has been further highlighted at the end of the literature review section in lines 440-443. 

“…In particular, there is an evident lack of empirical research for the aforementioned 4 country categories based on national income. Also, there were limited research available in the context of transitional economies and LDCs. The current study contributes to this gap in the existing literature.”

Furthermore, in the revised version, we have included a new section on the theoretical framework from the lines 159-196. 

“The relationship between REC, NREC, economic growth, and CO2 emissions is explained by 3 theoretical frameworks namely, (1) the 3Ps framework, (2) Natural Resource Curse and (3) the Porter Hypothesis. (1) The 3Ps framework is a common concept in the disciplines of sustainability and corporate social responsibility. The 3Ps stand for People, Planet, and Profit. These three factors must be taken into consideration by businesses while implementing sustainable practices. Planet and profit from the aforementioned three factors have been used in this study. Due to their contribution to reduced greenhouse gas emissions and lower detrimental effects on natural ecosystems, renewable energy sources have a lower environmental impact than conventional fossil fuels. Lowering CO2 emissions is crucial for decreasing effects of climate change, such as sea level rise, more frequent and severe weather, and other detrimental effects on the ecosystem. These components fall within the concept of ‘planet’. Long term energy cost savings, improved brand reputation, and access to new markets that value sustainability can all be achieved by businesses using renewable energy. Economic growth can also help businesses by raising consumer demand for goods and services, but to ensure sustainability over the long term, it must be in harmony with sustainability considerations. These elements can be considered as the ‘profit’ [20]. (2) Natural resource curse is the theory which highlights the difficulties that countries confront in managing their natural resources. The development of renewable energy sources also offers a theoretical framework for tackling these issues and fostering economic growth. Diversifying the economy to minimise the reliance on natural resources is one strategy for overcoming the resource curse. By investing in energy alternatives that don't rely on non-renewable resources, renewable energy gives nations the opportunity to accomplish these objectives. This can also reduce dependency on fossil fuels exports and result in the development of new businesses and jobs [21]. According to (3) the Porter Hypothesis, environmental laws can encourage innovation, competitiveness, and economic growth. This concept suggests that environmental rules encourage companies to develop more effective and environmentally friendly production methods by opening avenues for new markets for clean technologies. For businesses that use cleaner technologies, this enables cost savings, better profitability, and increased competitiveness. According to the Porter Hypothesis, environmental restrictions can open up new markets for renewable energy technology, lowering reliance on non-renewable energy sources and preventing climate change. This is relevant to renewable energy. Environmental laws can also boost the economy by enhancing public health, decreasing healthcare expenses, and boosting productivity [22]. This study analysed the estimation results in the context of these well-accepted theories to illustrate the link between economic growth, energy consumption and CO2 emissions.”

Reviewer 1 comment 6: The author(s) need to present the flow of the analysis. The author(s) can incorporate this in the data and method section.

Authors’ Response to Reviewer 1 comment 6: Many thanks. We agree with your suggestion. In the revised version of the paper, we provide a new section detailing the flow of analysis before the Empirical Results and Discussion section in lines 517-522 as below.

“As the initial step to the analysis, descriptive analysis, correlation analysis and IRF graphs were conducted. Granger-causality analysis was tested as the second stage of the analysis. Before accomplishing the Granger-causality test, this study conducted the first- and second-generation unit root tests. Afterwards lag selection criteria was carried out. Then, stability condition was tested, and finally Granger-causality test was performed.”

Reviewer 1 comment 7: The discussion section needs minor improvement. This can be done by comparing your results with prior studies. Kindly compare your study with the following studies

https://doi.org/10.1080/13504509.2022.2115577

https://doi.org/10.1177/0958305X231167463

Authors’ Response to Reviewer 1 comment 7: Thank you for these suggestions. 

We have addressed this comment by comparing results with prior studies in the results and discussion section of the revised version of the paper. Also, the proposed articles were incorporated into the discussion of results. Please refer to the revised version of the paper in lines 609-614.

“..However, our results are different to the study which found that economic growth reduce CO2 emissions in both short and long-term in Nordic countries [36]. Furthermore, another study found that long-run dynamic connections exist among industrial CO2 emissions and oil price [85]…”

Reviewer 1 comment 8: Please insert graphical presentations for your results.

Authors’ Response to Reviewer 1 comment 8: Thank you for your suggestion. In addition to the various graphical presentation of our results, such as box plots, violin graphs to identify descriptive results and IRF graphs to identify shocks along with the timeline in the paper, we have provided S5 and S6 Appendices with field maps.

Field maps are used for a better understanding and presenting view of the granger causality results using a world map. The mentioned appendices, S5 and S6, are quoted as follows in the revised manuscript.

Please refer to the revised version of the paper lines 785-787 and 814-815.

“…S5 Appendix demonstrates the Granger-causality results on REC, NREC and GDP for each country category by using the field maps.”

“…The Granger-causality results on CO2 and GDP for each country category are illustrated in S6 Appendix.” 

Reviewer 1 comment 9: Use of English language need slight improvement. Revise almost all sentences in the manuscript with appropriate use of grammar, punctuation, and speech.

Authors’ Response to Reviewer 1 comment 9: Noted with many thanks. The paper has been revised thoroughly and in-depth copy edit has conducted by an experienced copyeditor. We trust that the revised manuscript is free from any language, grammar, and punctuation errors.

We appreciate you for your major effort on this article. 

Reviewer 2 General comment: The study aims to examine the relationship between renewable and non-renewable energy use, CO2 emissions and economic growth in developed, developing, and Least-developed Economies in Transition from 1990 to 2019.

The manuscript suggests that the study included 152 countries and used Granger-causality as a methodology to investigate the links between the variables. The findings of the study indicated that there may or may not be a relationship between the consumption of renewable and non-renewable energy sources and economic growth.

Furthermore, there was no Granger-causal relationship between GDP and renewable energy consumption outside of Economies in Transition, and the GDP and CO2 of all countries had a one-way relationship.

Authors’ Response to Reviewer 2 general comments: Thank you for your positive remarks on our paper.

Reviewer 2 comment 1: While the manuscript is organized and well-written with chapters that follow a scientific study, the methodology used in the study could be improved. The reviewer suggests that the authors use additional econometrics tests to support the discussions and conclusions. The study's great literature review and inclusion of sufficient studies on the topic are positives. However, the findings do not contribute significantly to the literature.

Authors’ Response to Reviewer 2 comment 1: Thank you for highlighting this point. We have addressed your concern in the revised paper in several ways.

Firstly, the methodology has been further improved by graphical presentations as presented in S5 and S6 Appendix.

Secondly, second-generation unit root test was employed and explained in lines 743-745 as follows.

“Furthermore, second generation unit root test introduced by [88] was conducted in advance. The result for the test is shown in S3 Appendix and the final conclusion of the results are the same as in the LLC unit root test.”

Thirdly, sensitivity analysis is done for the study and presented in S7 Appendix which is mentioned in lines 825-829. 

“Furthermore, in line with a previous study [93], lag-based sensitivity analysis was conducted to explore whether results are insensitive to changes in the model. Results for the lag-based analysis presented in S7 Appendix. However, according to the lag selection process for the current study, lag 1 is the most suitable for the analysis and the analysis was conducted thereon.”

Finally, we have carefully integrated our findings with prior studies to highlight the differences and similarities of our findings and explaining how our analysis contribute to the existing literature (please see lines 602-603, 769-771, and 774-778 in the revised version of the paper).

“…..This could be as a result of the perfect correlation between REC and NREC data. However, these results differ from [58] in the BRICST, which demonstrated the positive effect of economic growth on renewable energy usage.…”

“……The study based on the BRICS region also revealed that the NREC, and economic growth had a bi-directional causative link with the environmental impact, but not with REC [50].…”

“…Similarly, our results are also inconsistent with a study [25], which disclosed a uni-directional causal link between economic growth and energy utilize in the top 5 carbon emitting countries (China, the United States, India, Brazil, and Russia). ….”

Reviewer 2 comment 2: The study highlights that it is crucial to take action to lower CO2 emissions, which would develop renewable energy while also stimulating the economy. While the study provides an analysis of the variables discussed in the four subgroups, there is a need for more empirical studies that encompass all regions.

Authors’ Response to Reviewer 2 comment 2: Thank you for the valuable comment. We have addressed your suggestion by incorporating several new studies into our analysis in the revised version of the paper. Please see lines 324-326, 344-350, 362-366, and 366-369.

“…Some studies’ primary suggestions include decreasing fossil fuel subsidies and increasing investment in renewable energy [47].….”

“….On the one hand, a study conducted for the E7 countries (Brazil, China, India, Indonesia, Mexico, Russia, and Turkey) between 1990 and 2015 proved that using renewable energy improves environmental quality by reducing both ecological footprint and CO2 emissions. On the other hand, using non-renewable energy increases both the environmental impact and CO2 emissions. The results of the causality test suggest that there is a uni-directional causal relationship running from NREC to economic growth [52].….”

“…Similarly, in China, CO2 emissions are positively influenced by economic growth, coal consumption, and natural resources at all frequencies, or both in the short and long-run, leading to environmental degradation, using the Wavelet Local Multiple Correlation bivariate cases [56].…”

“….According to the study that utilised quantile-on-quantile regression (QQR) method, Nigeria and Mexico experienced positive effects of energy resources on economic growth in the majority of quantiles [57].….”

Reviewer 2 comment 3: In conclusion, the manuscript is partly technically sound but needs improvement in methodology, particularly in the use of econometric techniques. I would suggest the used second-generation unit-root testing and better causality methodology.

Authors’ Response to Reviewer 2 comment 3: Thank you for your suggestions and feedback. As recommended, we have used the second-generation unit root test in the revised manuscript. The results of the second-generation unit root test are consistent with the LLC unit root test, which we have discussed in lines 743-745 as,

“Furthermore, second generation unit root test introduced by [88] was conducted in advance. The result for the test is shown in S3 Appendix and the final conclusion of the results are the same as in the LLC unit root test.”

However, this study used “pvargranger” command to get the results for the causal relationship between variables. For each equation, the postestimation function “pvargranger” runs Granger causality Wald tests. By default, “pvargranger” utilizes the most recent (active) results. Therefore, as the opinion of the authors this study used a better causality methodology. Furthermore, we have discussed the use of Granger-causality in lines 502-515 as, 

“In the field of causal analysis, the use of Granger-causality offers several benefits and insights. Firstly, it allows us to identify causal interference, which involves examining whether one time series can effectively forecast another. By establishing such causal relationships, the accuracy of forecasting processes can be significantly enhanced, leading to more reliable predictions. Secondly, Granger-causality aids in determining the direction of causality. This is crucial for researchers as it helps identify which variable serves as the leading factor and which one lags behind in a causal relationship. This understanding of directionality enables the identification of key variables and their relationships, providing valuable insights for both researchers and policy makers in formulating appropriate policies. Furthermore, Granger-causality analysis provides a measure of the strength of the causal relationship at selected significance levels. This information empowers researchers and policy makers to evaluate and assess the reliability of the obtained results before making informed decisions and taking appropriate actions. Therefore, it can be assumed that this study using Granger-causality to identify the causal relationships, will add value to the existing literature.”

Thank you for your instructive contribution on this article to be success. 

Reviewer 2 comment 4: The findings of the study do not contribute significantly to the literature, but the study has a great literature review, and sufficient studies are provided on the topic. The authors are advised to review the econometric techniques used for the study as there are better techniques available.

Authors’ Response to Reviewer 2 comment 4: Thank you for drawing our consideration to that. The implemented adjustment with past research that add considerably to the literature, as requested, is provided in revised manuscript. 

A second-generation unit root test is also included in the research line 743-745 as,

“Furthermore, second generation unit root test introduced by [88] was conducted in advance. The result for the test is shown in S3 Appendix and the final conclusion of the results are the same as in the LLC unit root test.”

Thank you for drawing our consideration to that. The implemented adjustment with past research that add considerably to the literature, as requested, is provided in revised manuscript. 

A second-generation unit root test is also included in the research line 743-745 as,

“Furthermore, second generation unit root test introduced by [88] was conducted in advance. The result for the test is shown in S3 Appendix and the final conclusion of the results are the same as in the LLC unit root test.”

Additionally, as illustrated in S5 and S6 Appendix, field maps are used for a better understanding and presenting view of the granger causality results using a world map. The current manuscript was revised in lines 785-789 and 814-815 as, 

“……S5 Appendix demonstrates the Granger-causality results on REC, NREC and GDP for each country category by using the field maps.” 

“……The Granger-causality results on CO2 and GDP for each country category are illustrated in S6 Appendix.”

Thank you for your major effort on this article. 

Reviewer 3 comment 1: The novelty of paper is extremely weak. The authors are suggested to make the manuscript novel in terms of methodology (atleast). Otherwise, the manuscript seems missfit for publication.

Authors’ Response to Reviewer 3 comment 1: Many thanks for your suggestion. We have improved the methodology of the study in the revised version of the paper in several ways. 

Firstly, the methodology has been further improved by graphical presentations as presented in S5 and S6 Appendix.

Secondly, the second-generation unit root test was employed and explained in lines 743-745 as follows.

“Furthermore, second generation unit root test introduced by [88] was conducted in advance. The result for the test is shown in S3 Appendix and the final conclusion of the results are the same as in the LLC unit root test.”

Thirdly, sensitivity analysis is done for the study and presented in S7 Appendix which is mentioned in lines 825-829. 

“Furthermore, in line with a previous study [93], lag-based sensitivity analysis was conducted to explore whether results are insensitive to changes in the model. Results for the lag-based analysis presented in S7 Appendix. However, according to the lag selection process for the current study, lag 1 is the most suitable for the analysis and the analysis was conducted thereon.”

Reviewer 3 comment 2: The literature needs to be significantly improved. The authors can use these studies to update the literature review section:

1. Monetary and energy policy interlinkages: The case of renewable energy in the US; 

2. Investigating the Impact of Climate Change Mitigation Technology on the Transport Sector CO2 Emissions: Evidence from Panel Quantile Regression. 

3. Does sectoral energy consumption depend on trade, monetary, and fiscal policy uncertainty? Policy recommendations using novel bootstrap ARDL approach. 

4. Exploring the dynamic relationship between energy efficiency, trade, economic growth, and CO2 emissions: Evidence from novel fourier ARDL approach. 

5. Does green environmental policy promote renewable energy consumption in BRICST? Fresh insights from panel quantile regression.

6. Do geopolitical risk and energy consumption contribute to environmental degradation? Evidence from E7 countries.

7. Applying a dynamic ARDL approach to the Environmental Phillips Curve (EPC) hypothesis amid monetary, fiscal, and trade policy uncertainty in the USA. 

8. Investigating the Environmental Kuznets Curve hypothesis amidst geopolitical risk: Global evidence using bootstrap ARDL approach. 

9. Do economic policy uncertainty and geopolitical risk surge CO2 emissions? New insights from panel quantile regression approach.

10. Impact of economic policy uncertainty on CO2 emissions in the US: Evidence from bootstrap ARDL approach. 

11. Impact of economic policy uncertainty on CO2 emissions: evidence from top ten carbon emitter countries.

Authors’ Response to Reviewer 3 comment 2: Thank you for your insightful comment. We have further extended the literature by incorporating the suggested articles in lines 85-86, 86-92, 275 -286, and 344-350.

“…..Furthermore, it has been found that energy consumption results in greater emissions in both short and long term globally [8].….”

“……In addition, the scientific literature provides mixed findings on the nexus between Renewable Energy Consumption (REC) and economic growth, owing to different methodological approaches, types of data used and time period under consideration. For example, Gross Domestic Product (GDP) per capita, renewable energy, and non-renewable energy have different effects on CO2 emissions in BRICST (Brazil, Russia, India, China, South Africa and Turkey) countries [9].…..”

“……As per the research conducted in the USA using the bootstrap ARDL method, energy consumption negatively impacts monetary policy uncertainty between 1990M1 and 2020M12 [32]. By utilizing the unit root test, it turned out to be that expansionary monetary policy positively impacted the REC during the long-run, and short-run, and inversely in the USA [33]. Similarly, in another study based on the USA, the dynamic ARDL model’s findings show that while fiscal policy uncertainty reduces emissions, monetary policy uncertainty increases emissions for both short and long- run. Last but not least, trade policy uncertainty has no impact on CO2 emissions [34]. Furthermore, according to the bootstrap ARDL technique results in the USA, excessive EPU may be accountable for short term environmental degradation because it increases CO2 emissions. In contrast, EPU reduces CO2 emissions in the long-run, suggesting that high EPU improves environmental quality in the long-run [35].…”

“…..On the one hand, a study conducted for the E7 countries (Brazil, China, India, Indonesia, Mexico, Russia, and Turkey) between 1990 and 2015 proved that using renewable energy improves environmental quality by reducing both ecological footprint and CO2 emissions. On the other hand, using non-renewable energy increases both the environmental impact and CO2 emissions. The results of the causality test suggest that there is a uni-directional causal relationship running from NREC to economic growth [52].…..”

Reviewer 3 comment 3: The motivation behind the employed methodology seems weak. The authors should also work on it

Authors’ Response to Reviewer 3 comment 3: Thank you very much for your insightful comment. In this study, we have used Granger-causality approach for the analysis because our primary objective to test and determine the causal link between GDP, REC, NREC and CO2. We have discussed the use of Granger-causality in lines 502- 515 to acknowledge the motivation behind the employed methodology.

“In the field of causal analysis, the use of Granger-causality offers several benefits and insights. Firstly, it allows us to identify causal interference, which involves examining whether one time series can effectively forecast another. By establishing such causal relationships, the accuracy of forecasting processes can be significantly enhanced, leading to more reliable predictions. Secondly, Granger-causality aids in determining the direction of causality. This is crucial for researchers as it helps identify which variable serves as the leading factor and which one lags behind in a causal relationship. This understanding of directionality enables the identification of key variables and their relationships, providing valuable insights for both researchers and policy makers in formulating appropriate policies. Furthermore, Granger-causality analysis provides a measure of the strength of the causal relationship at selected significance levels. This information empowers researchers and policy makers to evaluate and assess the reliability of the obtained results before making informed decisions and taking appropriate actions. Therefore, it can be assumed that this study using Granger-causality to identify the causal relationships, will add value to the existing literature.”

Reviewer 3 comment 4: There is a need to perform sensitivity analysis to explore whether results are insensitive to changes in model/methodology/data.

Authors’ Response to Reviewer 3 comment 4: Many thanks for this suggestion. To address your comment, we carried out a lag-based sensitivity analysis which has been presented in S7 Appendix. The lag 1 was found to be the best fit for the model according to the lag selection process and the analysis was conducted thereon. Please see lines 825-829 in the revised manuscript which is quoted as follows. 

“Furthermore, in line with a previous study [93], lag-based sensitivity analysis was conducted to explore whether results are insensitive to changes in the model. Results for the lag-based analysis presented in S7 Appendix. However, according to the lag selection process for the current study, lag 1 is the most suitable for the analysis and the analysis was conducted thereon.”

Reviewer 3 comment 5: Results should be explained in a detailed manner

Authors’ Response to Reviewer 3 comment 5: Thank you for your comment. We have addressed this comment in the revised manuscript in several ways. 

Firstly, the Descriptive statistics section in lines 531-534 was expanded as follows. 

“…Descriptive statistics show that the highest average GDP growth as 4.02% and highest average REC as 73% in LDCs. Furthermore, developed countries shows the highest average CO2 emissions indicating that developed countries contribute more proportion of CO2 to the world…”

Secondly, the discussion of Granger-causality results was further expanded as follows in lines 733-734 and 749-750.

“…All the variables are significant at the level of 1% for each country category as well as in the global level and the data set is stationary…”

“…Lag 1 was chosen for the rest of the analysis…” 

Thirdly, we have addressed several new studies into our analysis in the revised version of the paper. Please see lines 324-326, 344-350, 362-366, and 366-369.

“…Some studies’ primary suggestions include decreasing fossil fuel subsidies and increasing investment in renewable energy [47].….”

“….On the one hand, a study conducted for the E7 countries (Brazil, China, India, Indonesia, Mexico, Russia, and Turkey) between 1990 and 2015 proved that using renewable energy improves environmental quality by reducing both ecological footprint and CO2 emissions. On the other hand, using non-renewable energy increases both the environmental impact and CO2 emissions. The results of the causality test suggest that there is a uni-directional causal relationship running from NREC to economic growth [52].….”

“…Similarly, in China, CO2 emissions are positively influenced by economic growth, coal consumption, and natural resources at all frequencies, or both in the short and long-run, leading to environmental degradation, using the Wavelet Local Multiple Correlation bivariate cases [56].…”

“….According to the study that utilised quantile-on-quantile regression (QQR) method, Nigeria and Mexico experienced positive effects of energy resources on economic growth in the majority of quantiles [57].….”

Reviewer 3 comment 6: policy implications should be in line with the main findings

Authors’ Response to Reviewer 3 comment 6: We have noted your comment with many thanks. We have improved the policy implication section which are aligned with the main findings of the study in lines 888-921 in the revised version of the paper.

“The findings of this study provide useful policy insights in the areas of REC, NREC, GDP growth and CO2 emissions, which are summarised below.

 Based on this analysis, the main recommendation is that all countries, particularly the developed countries need to take necessary initiative to reduce overall CO2 levels to achieve sustainable economic growth. Majority of the developed countries have higher GDP levels and production capacities which may lead to increased energy consumption from non-renewable energy sources compared to other nations. Policymakers should focus more on adopting renewable energy sources to produce energy as this may contribute to lower CO2 emissions. All the energy related investments and developments in developed countries should give priority to and include renewable energy as a key performance indicator in their investment appraisal considerations. 

Similar concerns are applicable in the context of transitional economies. Nevertheless, policymakers should focus on both GDP and CO2 when making decisions because this study’s results indicated a bi-directional causality between the two variables in transitional economies. Hence, policy decisions related to sustainable economic growth in transitional economies should be carefully analysed before implementing due to this causal relationship. Therefore, to mitigate the adverse environmental implications of increased NREC, it is worthwhile for transitional economies to invest more on renewable energy sources and implement strategies to reduce CO2 emissions.

Ultimately, considering the overall benefits, it is advisable for all the nations to go for more renewable energy sources for a sustainable future. CO2 emission is one of the most pressing environmental issues nowadays and increasing cleaner and REC will have a direct considerable impact on reducing CO2 emissions. All the nations should contribute to the world economy by setting up clear pathways to achieve sustainable economic goals. In general, the findings of this current study imply that it is preferable to reduce the CO2 emissions by implementing necessary policies while improving GDP in all countries across the globe.”

Thank you for your valuable input.

Reviewer 4 comment 1: The Abstract must report the aim of the study, the basic information on the sample (time span, countries analysed), the empirical methodology used, the main findings, and the relevant policy implications.

Authors’ Response to Reviewer 4 comment 1: Noted with many thanks for the valuable comment. The abstract has been revised by including the requested information in lines 35-39, 44-50, and 52-53.

“…The study has examined the links between renewable and non-renewable energy use, CO2 emissions and economic growth in developed, developing, and LDCs and Economies in Transition between 1990 and 2019 in152 countries. Granger-causality has been used as the methodology to investigate the link between the variables ...”

“…The results indicated that no Granger-causal relationship exists between GDP and REC outside of Economies in Transition. Additionally, the GDP and CO2 of all countries have a one-way relationship. Nevertheless, research indicates that GDP and CO2 have a bi-directional link in Economies in Transition, a uni-directional relationship in developing countries, and no meaningful association in developed and LDCs…”

“…Ultimately, more nations should choose renewable energy sources to build a more sustainable future…”

Reviewer 4 comment 2: The Introduction should highlight the relevance of the topic, the novelty of the results, the importance of policy implications, the sample’s choice, the methodology’s appropriateness, the data used, the contribution to the literature, and the limitations of the study.

Authors’ Response to Reviewer 4 comment 2: Thank you very much for highlighting this comment. We have further strengthened the introduction in lines 130-133 by including suggested information. 

“…However, this study does not address the creation of jobs, improved quality of life in rural areas, improved public health as a result of reduced pollution, or increased public and professional awareness; it is because the study focused only on renewable energy, environmental sustainability and economic growth…”

Reviewer 4 comment 3: The literature review is partial and incomplete, and some recent and relevant contributions should be cited and discussed. 

1. Role of technological innovation and globalization in BRICS economies: policy towards environmental sustainability. International Journal of Sustainable Development & World Ecology, 29(7), 593-610.

2. Can green resource productivity, renewable energy, and economic globalization drive the pursuit of carbon neutrality in the top energy transition economies? International Journal of Sustainable Development & World Ecology, 1-15.

3. Determinants of carbon emissions in Argentina: The roles of renewable energy consumption and globalization. Energy Reports, 7, 4747-4760.

4. Toward a sustainable environment and economic growth in BRICS economies: do innovation and globalization matter? Environmental Science and Pollution Research, 29(38), 57740-57757.

5. Does globalization moderate the effect of economic complexity on CO2 emissions? Evidence from the top 10 energy transition economies. Frontiers in Environmental Science, 555.

Testing the asymmetric effect of financial stability towards carbon neutrality target: The case of Iceland and global comparison. Gondwana Research.

Authors’ Response to Reviewer 4 comment 3: Thank you for your insightful comment. We have strengthened our literature review by incorporating the suggested articles in lines 335-339, 256-259, 359-362, 373-376, 313-316, and 289-291.

“…Additionally, a study on Brazil, Russia, India, China, and South Africa (BRICS) countries found that the regressors of technological innovation, globalization, non-renewable energy, and economic expansion have a bi-directional causal relationship with ecological footprint, according to the Dumitrescu-Hurlin panel causality test [50]…”

“…For the Granger-causality inference, a uni-directional causal interaction from economic growth to CO2 emissions is found in the top 10 energy transition economies (China, USA, UK, Germany, France, Japan, India, Spain, Brazil, and South Korea) [24]...”

“…The use of renewable energy and globalization have been shown to reduce emissions in Argentina, whereas the use of non-renewable energy has been found to increase emissions over the short and long-term according to a study based on the ARDL model [55]. …”

“…Another study on BRICS economies revealed that the use of renewable energy, exports, and technological advancements reduce consumption-based carbon emissions (CCO2), whereas economic expansion and imports increase CCO2 [59]…”

“…In the long-run, it has been identified that globalization, renewable energy, and economic complexity reduce carbon emissions in the top 10 energy transition economies (Iceland, France, New Zealand, UK, Finland, Austria, Switzerland, Denmark, Norway, and Sweden) [44]…”

“…Furthermore, energy use and income level are found to enhance CO2 emissions in Iceland using fully modified ordinary least squares regression and Dynamic Ordinary Least Square regression [37]…”

Reviewer 4 comment 4: The gap in literature should be reported in the literature section.

Authors’ Response to Reviewer 4 comment 4: Well noted with many thanks. The gap in literature has been further highlighted in lines in 440-443.

“...In particular, there is an evident lack of empirical research for the aforementioned four country categories based on national income. Also, there were limited research available in the context of transitional economies and LDCs. The current study contributes to this gap in the existing literature…”

Reviewer 4 comment 5: 

How the authors have derived the empirical model? There should be a thorough theoretical underpinning behind the model. This section should be based on the logic of the authors, and no citation/reference should appear here. This section will be followed by the empirical model.

Authors’ Response to Reviewer 4 comment 5: Well noted your comment with many thanks. We have included the theoretical framework as a separate section in lines 158-196 in the revised manuscript. 

“The relationship between REC, NREC, economic growth, and CO2 emissions is explained by three theoretical frameworks namely, (1) the 3Ps framework, (2) Natural Resource Curse and (3) the Porter Hypothesis. (1) The 3Ps framework is a common concept in the disciplines of sustainability and corporate social responsibility. The 3Ps stand for People, Planet, and Profit. These three factors must be taken into consideration by businesses while implementing sustainable practices. Planet and profit from the aforementioned three factors have been used in this study. Due to their contribution to reduced greenhouse gas emissions and lower detrimental effects on natural ecosystems, renewable energy sources have a lower environmental impact than conventional fossil fuels. Lowering CO2 emissions is crucial for decreasing effects of climate change, such as sea level rise, more frequent and severe weather, and other detrimental effects on the ecosystem. These components fall within the concept of ‘planet’. Long term energy cost savings, improved brand reputation, and access to new markets that value sustainability can all be achieved by businesses using renewable energy. Economic growth can also help businesses by raising consumer demand for goods and services, but to ensure sustainability over the long term, it must be in harmony with sustainability considerations. These elements can be considered as the ‘profit’ [20]. (2) Natural resource curse is the theory which highlights the difficulties that countries confront in managing their natural resources. The development of renewable energy sources also offers a theoretical framework for tackling these issues and fostering economic growth. Diversifying the economy to minimise the reliance on natural resources is one strategy for overcoming the resource curse. By investing in energy alternatives that don't rely on non-renewable resources, renewable energy gives nations the opportunity to accomplish these objectives. This can also reduce dependency on fossil fuels exports and result in the development of new businesses and jobs [21]. According to (3) the Porter Hypothesis, environmental laws can encourage innovation, competitiveness, and economic growth. This concept suggests that environmental rules encourage companies to develop more effective and environmentally friendly production methods by opening avenues for new markets for clean technologies. For businesses that use cleaner technologies, this enables cost savings, better profitability, and increased competitiveness. According to the Porter Hypothesis, environmental restrictions can open up new markets for renewable energy technology, lowering reliance on non-renewable energy sources and preventing climate change. This is relevant to renewable energy. Environmental laws can also boost the economy by enhancing public health, decreasing healthcare expenses, and boosting productivity [22]. This study analysed the estimation results in the context of these well-accepted theories to illustrate the link between economic growth, energy consumption and CO2 emissions.”

Reviewer 4 comment 6: Conclusion reiterates the results, which is completely undesirable. The authors should summarize the results within a maximum of 3 sentences. Moreover, the policies are completely vague, and it seems that the authors already had them in mind before starting the paper. The policies should be directly derived from the discussion of the results, and they should not go beyond the results.

Authors’ Response to Reviewer 4 comment 6: Thank you very much for notifying the comment and well noted. 

We have done the following to address your comment by inserting a new policy implications section in lines 888-921 as follows.

“The findings of this study provide useful policy insights in the areas of REC, NREC, GDP growth and CO2 emissions, which are summarised below.

 Based on this analysis, the main recommendation is that all countries, particularly the developed countries need to take necessary initiative to reduce overall CO2 levels to achieve sustainable economic growth. Majority of the developed countries have higher GDP levels and production capacities which may lead to increased energy consumption from non-renewable energy sources compared to other nations. Policymakers should focus more on adopting renewable energy sources to produce energy as this may contribute to lower CO2 emissions. All the energy related investments and developments in developed countries should give priority to and include renewable energy as a key performance indicator in their investment appraisal considerations. 

Similar concerns are applicable in the context of transitional economies. Nevertheless, policymakers should focus on both GDP and CO2 when making decisions because this study’s results indicated a bi-directional causality between the two variables in transitional economies. Hence, policy decisions related to sustainable economic growth in transitional economies should be carefully analysed before implementing due to this causal relationship. Therefore, to mitigate the adverse environmental implications of increased NREC, it is worthwhile for transitional economies to invest more on renewable energy sources and implement strategies to reduce CO2 emissions.

Ultimately, considering the overall benefits, it is advisable for all the nations to go for more renewable energy sources for a sustainable future. CO2 emission is one of the most pressing environmental issues nowadays and increasing cleaner and REC will have a direct considerable impact on reducing CO2 emissions. All the nations should contribute to the world economy by setting up clear pathways to achieve sustainable economic goals. In general, the findings of this current study imply that it is preferable to reduce the CO2 emissions by implementing necessary policies while improving GDP in all countries across the globe.”

Thank you for your worthwhile contribution on this article.

---

## [Decision Letter · Decision Letter 1]

8 Jun 2023

Nexus between Carbon Emissions, Energy Consumption, and Economic Growth: Evidence from Global Economies

PONE-D-23-08466R1

Dear Dr. Jayathilaka,

We’re pleased to inform you that your manuscript has been judged scientifically suitable for publication and will be formally accepted for publication once it meets all outstanding technical requirements.

Kind regards,

Roni Bhowmik, Ph.D.

Academic Editor

PLOS ONE

Additional Editor Comments (optional):

Reviewers' comments:

Reviewer's Responses to Questions

**Comments to the Author**

1. If the authors have adequately addressed your comments raised in a previous round of review and you feel that this manuscript is now acceptable for publication, you may indicate that here to bypass the “Comments to the Author” section, enter your conflict of interest statement in the “Confidential to Editor” section, and submit your "Accept" recommendation.

Reviewer #1: All comments have been addressed

Reviewer #2: All comments have been addressed

Reviewer #3: All comments have been addressed

Reviewer #4: All comments have been addressed

2. Is the manuscript technically sound, and do the data support the conclusions?

Reviewer #1: Yes

Reviewer #2: Yes

Reviewer #3: Yes

Reviewer #4: Yes

3. Has the statistical analysis been performed appropriately and rigorously? 

Reviewer #1: Yes

Reviewer #2: Yes

Reviewer #3: Yes

Reviewer #4: Yes

4. Have the authors made all data underlying the findings in their manuscript fully available?

Reviewer #1: Yes

Reviewer #2: Yes

Reviewer #3: Yes

Reviewer #4: Yes

5. Is the manuscript presented in an intelligible fashion and written in standard English?

Reviewer #1: Yes

Reviewer #2: Yes

Reviewer #3: Yes

Reviewer #4: Yes

6. Review Comments to the Author

Reviewer #1: Dear Authors

The corrections have been addressed. The manuscript can be published in it present form as it contribute significantly to the academic world. Welldone

Reviewer #2: Thank you for taking my comments into consideration and ammending the article as advised. I am satisified with the revised edition and recommend publication.

Reviewer #3: The authors made serious efforts to improve the manuscript. The commenst are addressed positively. The manuscript is now ready for publication.

Reviewer #4: Great job

This article is ready for publication

7. PLOS authors have the option to publish the peer review history of their article (what does this mean?). If published, this will include your full peer review and any attached files.

Reviewer #1: No

Reviewer #2: No

Reviewer #3: No

Reviewer #4: No

---

## [Editor Report · Acceptance letter]

14 Jun 2023

PONE-D-23-08466R1 

Nexus between Carbon Emissions, Energy Consumption, and Economic Growth: Evidence from Global Economies 

Dear Dr. Jayathilaka:

I'm pleased to inform you that your manuscript has been deemed suitable for publication in PLOS ONE. Congratulations! Your manuscript is now with our production department. 

Kind regards, 

on behalf of

Associate Professor Roni Bhowmik 

Academic Editor

PLOS ONE